# CYCle: Choosing Your Collaborators Wisely to Enhance Collaborative Fairness in Decentralized Learning

**Nurbek Tastan** *nurbek.tastan@mbzuai.ac.ae*
*Mohamed bin Zayed University of Artificial Intelligence (MBZUAI)*

**Samuel Horváth** *samuel.horvath@mbzuai.ac.ae*
*Mohamed bin Zayed University of Artificial Intelligence (MBZUAI)*

**Karthik Nandakumar** *nandakum@msu.edu*
*Michigan State University (MSU)*
*Mohamed bin Zayed University of Artificial Intelligence (MBZUAI)*

**Reviewed on OpenReview:** *https://openreview.net/forum?id=ygqNiLQqfH*

## Abstract

Collaborative learning (CL) enables multiple participants to jointly train machine learning (ML) models on decentralized data sources without raw data sharing. While the primary goal of CL is to maximize the expected accuracy gain for each participant, it is also important to ensure that the gains are **fairly** distributed: no client should be negatively impacted, and gains should reflect contributions. Most existing CL methods require central coordination and focus only on gain maximization, overlooking fairness. In this work, we first show that the existing measure of collaborative fairness based on the correlation between accuracy values without and with collaboration has drawbacks because it does not account for negative collaboration gain. We argue that maximizing mean collaboration gain (MCG) while simultaneously minimizing the collaboration gain spread (CGS) is a fairer alternative. Next, we propose the CYCle protocol that enables individual participants in a private decentralized learning (PDL) framework to achieve this objective through a novel reputation scoring method based on gradient alignment between the local cross-entropy and distillation losses. We further extend the CYCle protocol to operate on top of gossip-based decentralized algorithms such as Gossip-SGD. We also theoretically show that CYCle performs better than standard FedAvg in a two-client mean estimation setting under high heterogeneity. Empirical experiments demonstrate the effectiveness of the CYCle protocol to ensure positive and fair collaboration gain for all participants, even in cases where the data distributions of participants are highly skewed. The code can be found at `https://github.com/tnurbek/cycle`.

## 1 Introduction

Collaborative learning (CL) refers to a framework where several entities can work together by pooling their resources to achieve a common machine learning (ML) objective without sharing raw data. This approach offers particular advantages in domains like healthcare and finance that require access to extensive datasets, which can be difficult to acquire for any single entity due to the costs involved or privacy regulations such as HIPAA (Centers for Medicare & Medicaid Services, 1996) and GDPR (European Parliament & Council of the European Union, 2016; Albrecht, 2016). Federated learning (FL) (McMahan et al., 2017) is a specific case of CL that enables multiple entities to collectively train a shared model by sharing their respective model parameters or gradients with a central server for aggregation. However, FL methods can lead to information leakage associated with sharing gradients/parameter updates with an untrusted third-party server that can potentially carry out reconstruction attacks (Zhu et al., 2019; Zhao et al., 2020). Furthermore, most CL algorithms assume equal contributions from all participants. When this assumption is violated, there is

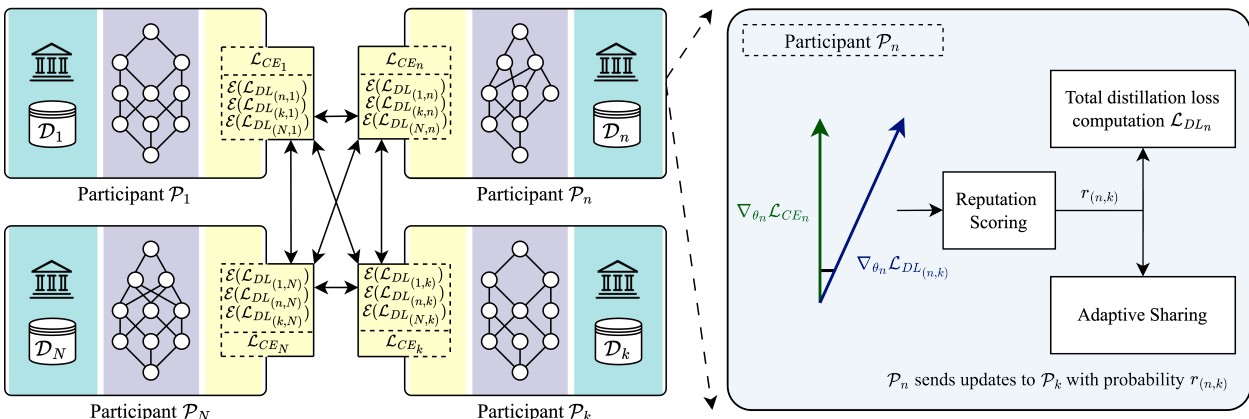

Figure 1: Illustration of the proposed Choose Your Collaborators Wisely (CYCle) protocol for Private Decentralized Learning (PDL).

little incentive for collaboration because the gains are not distributed fairly. To avoid these pitfalls, a fully decentralized learning algorithm that fairly rewards the contributions of individual participants is required.

Recent works have attempted to tackle the challenge of *collaborative fairness* (CF) in FL settings. Shapley value (SV) (Shapley et al., 1953) is a common choice for estimating participants' marginal utility, but its computation and communication costs are high. To mitigate this problem, variants of SV have been proposed in (Wang et al., 2020; Kumar et al., 2022; Xu et al., 2021; Tastan et al., 2024). But these methods only focus on FL with a central server. Existing private decentralized learning (PDL) algorithms such as CaPriDe learning (Tastan & Nandakumar, 2023), CaPC (Choquette-Choo et al., 2021), and Cronus (Chang et al., 2019) emphasize confidentiality, privacy, and utility, while ignoring collaborative fairness.

In this work, we aim to bridge this significant gap by designing the CYCle protocol for knowledge sharing in the PDL setting. We focus on three goals: maximizing the mean accuracy gain across all participants, ensuring that no participant suffers performance degradation due to collaboration, and that the accuracy gains are evenly distributed. To achieve these goals, we make the following contributions:

- We analyze the *collaborative fairness metric* based on the correlation coefficient between the contributions of participants and their respective final accuracies, and show that it fails in cases where the collaboration gain is negative.

- We introduce the *CYCle protocol* that regulates knowledge transfer among participants in a PDL framework based on their reputations to achieve better collaborative fairness. The proposed *reputation scoring* scheme uses gradient alignment between cross-entropy and distillation losses to accurately assess relative contributions made by collaborators in PDL.

- We extend the CYCle protocol to operate seamlessly on top of gossip-based decentralized algorithms such as Gossip-SGD, enabling fair collaboration.

- We theoretically study the CYCle protocol in the context of mean estimation between two participants and show that it outperforms FedAvg in the presence of data heterogeneity.

## 2 Related Work and Background

Several challenges associated with CL have been addressed recently, including privacy (Dwork et al., 2014; Choquette-Choo et al., 2021; Chang et al., 2019; Tastan & Nandakumar, 2023; Kairouz et al., 2021; Erlingsson et al., 2020; McMahan et al., 2018), confidentiality (Tastan & Nandakumar, 2023; Choquette-Choo et al., 2021; Chang et al., 2019), communication efficiency (McMahan et al., 2017), robustness (Lakshminarayanan et al., 2017; Athalye et al., 2018; Bagdasaryan et al., 2020), and fairness (Xu et al., 2021; Zhou et al.,

2021; Lyu et al., 2020). Most existing works focus only on FL algorithms with central orchestration such as FedAvg (McMahan et al., 2017). Due to the potential harm caused by gradient leakage in FL (Zhao et al., 2020; Zhu et al., 2019), which can result in the disclosure of sensitive user data, there is a growing interest in decentralized learning algorithms that do not require centralized orchestration. Focus has been mainly on ensuring confidentiality, privacy, and utility in decentralized learning (Tastan & Nandakumar, 2023; Choquette-Choo et al., 2021; Chang et al., 2019). While frameworks like Cronus (Chang et al., 2019) offer robustness and privacy in decentralized settings, they do not address fairness, which is central to our work. To our knowledge, CYCle is the first to explicitly target collaborative fairness in decentralized learning.

To achieve collaborative fairness in CL, it is critical to evaluate marginal contributions of participants (Jia et al., 2019; Xu et al., 2021; Shi et al., 2022; Jiang et al., 2023; Tastan et al., 2024; Donahue & Kleinberg, 2021). Because data distributions are often heterogeneous, some clients may contribute more valuable data, yet vanilla FL gives all participants the same global model, discouraging collaboration and hindering adoption. While Shapley Value (SV) can be used for data valuation, SV computation is expensive and hard to employ (Shapley et al., 1953). In CGSV (Xu et al., 2021), cosine similarity between local and global parameter updates is used to approximate the SV. The same cosine similarity approach is used in RFFL (Xu & Lyu, 2021) for reputation scoring. Kumar et al. (2022) used logistic regression models as proxies for client data and utilized an ensemble of them to approximate the SVs.

Compared to prior work that either assumes centralized orchestration or overlooks fairness, CYCle introduces a fully decentralized protocol that ensures fair collaboration by regulating knowledge transfer based on participant reputations. It departs from correlation-based fairness metrics that fail under negative collaboration gain and instead proposes a metric specifically designed for fair collaborative learning. CYCle extends naturally to Gossip-SGD, and we show both empirically and theoretically that it achieves better alignment between contribution and reward, particularly under data heterogeneity.

## 2.1 Preliminaries

**Notations.** Let $\tilde{\mathcal{M}}_\theta : \mathcal{X} \to \mathcal{Y}$ be a supervised classifier parameterized by $\theta$, where $\mathcal{X} \subseteq \mathbb{R}^d$ and $\mathcal{Y} = \{1, 2, \cdots, M\}$ denote the input and label spaces, respectively, $d$ is the input dimensionality, and $M$ is the number of classes. Let $\mathcal{M}_\theta : \mathcal{X} \to \mathcal{Z}$ denote a mapping from the input to the logits space ($\mathcal{Z} \subset \mathbb{R}^M$) and $\sigma_T$ be a softmax function (with temperature parameter $T$) that maps the logits into a probability distribution $\boldsymbol{p} = \sigma_T(\boldsymbol{z})$ over the $M$ classes. The sample is eventually assigned to the class with the highest probability. Given an input sample $\boldsymbol{x} \in \mathcal{X}$ and its ground truth label $y \in \mathcal{Y}$, let $\mathcal{L}_{CE}$ denote the cross-entropy (CE) loss based on the prediction $\sigma_T(\mathcal{M}_\theta(\boldsymbol{x}))$ and $y$. We denote $\mathcal{L}_{DL_{(i,j)}}$ as the pairwise distillation loss between two models $\mathcal{M}_{\theta_i}$ and $\mathcal{M}_{\theta_j}$. For example, the distillation loss (DL) could be computed as KL divergence between $\sigma_T(\mathcal{M}_{\theta_i}(\boldsymbol{x}))$ and $\sigma_T(\mathcal{M}_{\theta_j}(\boldsymbol{x}))$. Note that other types of distillation losses (Gou et al., 2021) can also be used in lieu of KL divergence.

**Decentralized Learning.** Let $N$ be the number of collaborating participants and $\mathcal{N} = \{1, 2, ..., N\}$. We assume that each participant $\mathcal{P}_n, n \in \mathcal{N}$ has its own local training dataset $\mathcal{D}_n$ and a hold-out validation set $\tilde{\mathcal{D}}_n$. Based on the local training set, each participant can learn a local ML model $\mathcal{M}_{\theta_n}$ by minimizing the local cross-entropy loss $\mathcal{L}_{CE_n}$. The validation accuracy of this standalone local model $\mathcal{M}_{\theta_n}$ on $\tilde{\mathcal{D}}_n$ is denoted as $\mathcal{B}_n$. The goal in a typical FL algorithm is to build a single global model $\mathcal{M}_{\theta_*}$ that has a better accuracy than the standalone accuracy of all the local models. In contrast, the goal of each participant in decentralized learning is to obtain a better local model $\mathcal{M}_{\theta_n^*}$, which has a higher local validation accuracy $\mathcal{A}_n$ after collaboration. In other words, each participant $\mathcal{P}_n$ aims to maximize its *collaborative gain* (CG), which is defined as $\mathcal{G}_n = (\mathcal{A}_n - \mathcal{B}_n)$. This is typically achieved by minimizing the following objective:

$$\mathcal{L}_n = \mathcal{L}_{CE_n} + \lambda_n \mathcal{L}_{DL_n}, \tag{1}$$

where $\mathcal{L}_{DL_n}$ denotes the total distillation loss of $\mathcal{P}_n$ and $\lambda_n > 0$ is a hyperparameter that controls the relative importance of the cross-entropy and distillation losses. Note that the term DL in the above formulation is used generically (without restricting to any specific loss function), and it intuitively captures the differences between the local model of a participant and the models of all its collaborators. The total DL of $\mathcal{P}_n$ can in

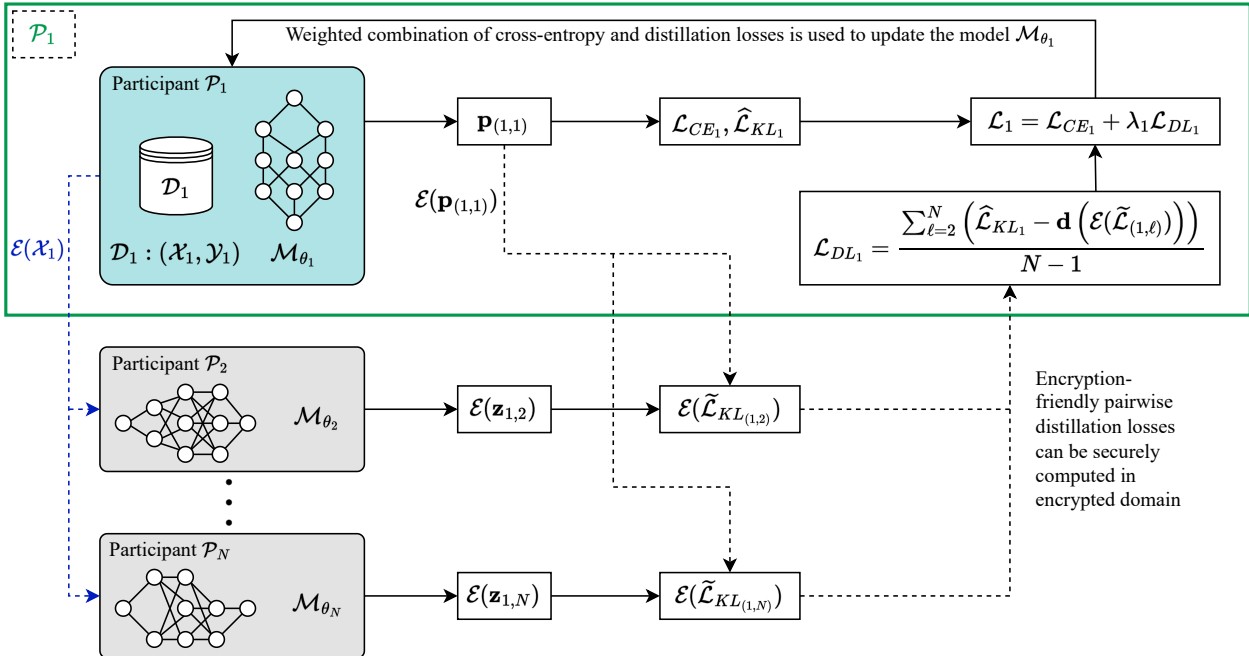

Figure 2: Illustration of confidential and private decentralized (CaPriDe) learning framework for $N$ participants, showing the learning process for only participant $\mathcal{P}_1$. Here, $\mathcal{D}_1 = (\mathcal{X}_1, \mathcal{Y}_1) = \{\boldsymbol{x}_{j,1}, y_{j,1}\}_{j=1}^{|\mathcal{D}_1|}$ is the local data of $\mathcal{P}_1$, $\mathcal{E}(\mathcal{X}_1)$ denotes the collection of encrypted unlabeled samples of $\mathcal{P}_1$, and $\boldsymbol{p}_{(1,l)}$ and $\boldsymbol{z}_{(1,l)}$ are the prediction probabilities and logits obtained by applying model $\mathcal{M}_{\theta_l}$ of participant $\mathcal{P}_l$ to $\mathcal{X}_1$. Black dashed line represents exchange of encrypted data between participants in each round, and blue dashed line denotes a single transfer at the beginning of the protocol. $\boldsymbol{d}$ is a decryption method. $\mathcal{L}_{CE}$ and $\mathcal{L}_{DL}$ represent the cross-entropy and distillation losses, respectively. In the context of our paper, $\mathcal{L}_{DL_{(1,k)}} := \widehat{\mathcal{L}}_{KL_1} - \boldsymbol{d}\left(\mathcal{E}\left(\widetilde{\mathcal{L}}_{KL_{(1,k)}}\right)\right)$.

turn be computed by aggregating the pairwise distillation losses as follows:

$$\mathcal{L}_{DL_n} = \sum_{k \in \mathcal{N} \setminus \{n\}} \lambda_{(n,k)} \mathcal{L}_{DL_{(n,k)}}, \tag{2}$$

where $\mathcal{L}_{DL_{(n,k)}}$ is the pairwise DL between $\mathcal{P}_n$ and $\mathcal{P}_k$, and $\lambda_{(n,k)}$ are the weights assigned to the pairwise losses between different participants.

**Private Decentralized Learning (PDL).** In the PDL framework, the total DL is computed in a privacy-preserving way without leaking the local data of participants. For example, CaPC learning (Choquette-Choo et al., 2021) leverages secure multi-party computation, homomorphic encryption, and differential privacy (DP) to securely estimate $\mathcal{L}_{DL_n}$. However, CaPC learning requires a semi-trusted third-party privacy guardian to privately aggregate local predictions, and the pairwise distillation losses are not accessible. In contrast, CaPriDe learning (Tastan & Nandakumar, 2023) utilizes fully homomorphic encryption (FHE) to securely compute an approximation of the pairwise DL losses. However, all existing PDL algorithms lack a mechanism to assess the contributions of different participants within the CL framework. Hence, $\lambda_{(n,k)}$ is usually set to $\frac{1}{(N-1)}$ and $\lambda_n = \lambda_0$, $\forall\, n \in \mathcal{N}$. $\lambda_0$ is typically determined through hyperparameter search. We refer to the above scenario as *vanilla PDL* (VPDL).

**CaPriDe Learning.** CaPriDe learning framework (Tastan & Nandakumar, 2023), as described above, uses a knowledge distillation approach and homomorphic encryption to enable secure and private computations. Figure 2 illustrates how CaPriDe learning works. Let $\mathcal{E}(\mathcal{X}_n) = \{\mathcal{E}(\boldsymbol{x}_{j,n})\}_{j=1}^{|\mathcal{D}_n|}$ be the collection of encrypted input samples of participant $\mathcal{P}_n$. Here, the encryption is based on an FHE scheme with the public key of $\mathcal{P}_n$. In CaPriDe learning, each party initiates the collaboration by publishing $\mathcal{E}(\mathcal{X}_n)$ to the other participants. Let $\boldsymbol{z}_{j,(n,k)}$ and $\boldsymbol{p}_{j,(n,k)}$ denote the logits vector and probability vector obtained when model $\mathcal{M}_{\theta_k}$ of $\mathcal{P}_k$ is applied

on the input sample $\boldsymbol{x}_{j,n}$, i.e., the $j^{th}$ training sample of $\mathcal{P}_n$. Assume that at round $(t+1), 0 \leq t < \tau$ of the collaboration, the current model of $\mathcal{P}_n$ is $\mathcal{M}_{\theta_n^t}$. Then, $\mathcal{P}_n$ performs one forward pass on its own training set $\mathcal{D}_n$ to obtain the predictions $\{\boldsymbol{p}_{j,(n,n)}\}_{j=1}^{|\mathcal{D}_n|}$. To enable knowledge transfer between participants, CaPriDe learning makes use of the knowledge distillation (KD) approach (Hinton et al., 2015), where a student model learns to mimic the predictions of a teacher model. In CaPriDe learning, there is no designated teacher model, and mutual KD between multiple peer models is used for knowledge transfer. Let $\mathcal{L}_{DL_{(n,k)}}$ denote the pairwise distillation loss between predictions of $\mathcal{P}_n$ and $\mathcal{P}_k$, which is defined as:

$$\mathcal{L}_{DL_{(n,k)}} = \sum_{j=1}^{|\mathcal{D}_n|} D_{DL}\left(\boldsymbol{p}_{j,(n,n)}, \boldsymbol{p}_{j,(n,k)}\right) = \widehat{\mathcal{L}}_{KL_n} - \boldsymbol{d}\left(\mathcal{E}\left(\widetilde{\mathcal{L}}_{KL_{(n,k)}}\right)\right) \tag{3}$$

where $D_{DL}(\boldsymbol{p}_1, \boldsymbol{p}_2)$ denotes the mimic distance between two predictions $\boldsymbol{p}_1$ and $\boldsymbol{p}_2$ and

$$\widehat{\mathcal{L}}_{KL_n} = \sum_{j=1}^{|\mathcal{D}_n|}\left(\boldsymbol{p}_{j,(n,n)} \cdot \log \boldsymbol{p}_{j,(n,n)}\right), \qquad \mathcal{E}\left(\widetilde{\mathcal{L}}_{KL_{(n,k)}}\right) = \sum_{j=1}^{|\mathcal{D}_n|} \mathcal{E}\left(\boldsymbol{p}_{j,(n,n)}\right) \cdot \frac{\mathcal{E}\left(\boldsymbol{z}_{j,(n,k)}\right)}{T}, \tag{4}$$

where $T$ is a temperature parameter and $\widetilde{\mathcal{L}}_{KL_{(n,k)}}$ is encryption-friendly, approximate KL divergence as per CaPriDe learning (Tastan & Nandakumar, 2023).

## 3 Towards a Better Fairness Metric

In (Lyu et al., 2020; Xu & Lyu, 2021), collaborative fairness (CF) is defined as follows:

**Definition 1** (Collaborative Fairness). In a federated system, a high-contribution participant should be rewarded with a better performing local model than a low-contribution participant. Mathematically, fairness can be quantified by the correlation coefficient between the contributions of the participants and their respective final model accuracies.

In the above CF Definition 1, the contribution of a participant is typically evaluated based on its standalone local accuracy ($\mathcal{B}_n$). Thus, the CF metric attempts to ensure that participants with higher standalone accuracy receive a model with higher final accuracy ($\mathcal{A}_n$) compared to that received by other participants. However, the problem with this approach can be illustrated using a simple example. Consider two participants $\mathcal{P}_1$ and $\mathcal{P}_2$ with standalone accuracy of 60% and 80%, respectively. Suppose that after collaboration, $\mathcal{P}_1$ ($\mathcal{P}_2$) receives a model with an accuracy of 70% (70.1%). In this scenario, the system is considered perfectly fair (with a fairness measure of 1) according to the above CF definition, because $\mathcal{P}_2$ receives a model with higher accuracy compared to $\mathcal{P}_1$. However, it is obvious that only $\mathcal{P}_1$ benefits from this collaboration and achieves a CG of $\mathcal{G}_1 = 10\%$, whereas $\mathcal{P}_2$ gains nothing. Strictly speaking, $\mathcal{P}_1$ has a negative CG of $\mathcal{G}_2 = -9.9\%$, but it can always dump the collaboratively trained model and use its own standalone model to achieve zero CG. This scenario cannot be considered fair, because the participant with the lower contribution benefits significantly without helping the participant with a higher contribution.

The problem with CF quantification in Definition 1 is that it overlooks the scenario where the collaboration gain is negative. This is usually not the case in FL with iid settings, because all the participants have similar standalone accuracy, and the accuracy of the collaboratively learned model is much higher than any of the individual standalone accuracies. However, in extreme non-iid settings (which is where collaborative fairness is most needed!), the problem of negative CG can be encountered often. Hence, there is a need for a better metric to quantify collaborative fairness.

One possible solution is to measure the rewards in terms of CG rather than final model accuracy. Fairness is then quantified by the correlation between participant contributions ($\mathcal{B}_n$) and their CGs ($\mathcal{G}_n$). However, this metric is problematic: participants with high standalone accuracy are harder to improve. Going back to the earlier example, if $\mathcal{P}_1$ improves from 60% to 70%, $\mathcal{P}_2$ has to improve its accuracy from 80% to more than 90% to achieve a fairness score of 1. While it is much easier to satisfy the first condition, it is hard to meet the second condition using any CL algorithm. Hence, the fairness metric based on the correlation between standalone accuracy and CG is likely to be negative for most cases, indicating poor fairness.

A better alternative is to minimize the probability that the CG is negative or zero, i.e., $P(\mathcal{G}_n \leq 0)$ should be small. Let $\mu = \mathbb{E}[\mathcal{G}_n]$ and $\nu = \sqrt{\mathrm{Var}[\mathcal{G}_n]}$ denote the mean and standard deviation of the collaboration gains, respectively. Intuitively, this is possible only when $\mu$ is a large positive value, and if $\nu$ is small. Mathematically, using the well-known Chebyshev single-tail inequality, we can show that $P(\mathcal{G}_n \leq 0)$ is bounded by $\frac{\nu^2}{\nu^2+\mu^2} = \frac{1}{(1+(\mu/\nu)^2)}$, provided $\mu > 0$. Thus, minimizing $P(\mathcal{G}_n \leq 0)$ requires maximizing $\mu$ and minimizing $\nu$. Let the mean collaboration gain (MCG) and the collaboration gain spread (CGS) be defined as the sample mean and standard deviation of CG across participants, i.e.,

$$\mathrm{MCG} = \frac{1}{N} \sum_{n=1}^{N} \mathcal{G}_n, \qquad \mathrm{CGS} = \sqrt{\frac{1}{(N-1)} \sum_{n=1}^{N} (\mathcal{G}_n - \mathrm{MCG})^2}. \tag{5}$$

Hence, an ideal decentralized learning scheme should maximize MCG (achieve better utility), while at the same time minimize CGS (ensure that all participants benefit fairly). This minimizes the likelihood of a participant being negatively impacted by the collaboration (having a negative CG).

## 4 Proposed CYCle Protocol

**Assumptions and Scope.** In this work, we restrict ourselves to the PDL framework presented in Eqs. 1 and 2. We also assume a cross-silo setting, where the number of participants is relatively small ($N < 10$) to ensure the practical feasibility of PDL. Our method requires a PDL algorithm that can compute pairwise DL in a privacy-preserving way. While we employ CaPriDe learning (Tastan & Nandakumar, 2023) as the baseline method for PDL in this work, other alternatives can also be used. Complete description of the baseline PDL method and discussions about its convergence, communication efficiency, or privacy guarantees are beyond the scope of this work.

**Rationale.** To maximize its individual CG (and hence maximize MCG), each participant must intuitively give more importance to distillation losses corresponding to other reliable participants, while de-emphasizing distillation losses corresponding to unreliable participants. To minimize CGS, each participant must penalize other unreliable participants (who do not contribute to their own learning) by sharing less knowledge with them. To achieve both

---

**Algorithm 1** CYCle $\left(t, k, \nabla_{\theta_n} \mathcal{L}_{CE_n}\right)$

**Input**: Momentum factor $\alpha$
1: $\phi \leftarrow$ Is Update Available from $\mathcal{P}_k$?
2: **Reputation Scoring**
3: **if** $\phi = 0$ **then**
4: $\quad \mathcal{L}_{DL_{(n,k)}} \leftarrow 0$
5: **else**
6: $\quad \mathcal{L}_{DL_{(n,k)}} \leftarrow$ Get DL from $\mathcal{P}_k$
7: $\quad$ **if** $(t \mod R) = 0$ **then**
8: $\quad\quad s \leftarrow \dfrac{1 - \cos(\nabla_{\theta_n}\mathcal{L}_{CE_n}, \nabla_{\theta_n}\mathcal{L}_{DL(n,k)})}{2}$
9: $\quad\quad \tilde{r} \leftarrow h(s)$
10: $\quad\quad r_{(n,k)}^{t} \leftarrow \alpha r_{(n,k)}^{t-1} + (1-\alpha)\tilde{r}$
11: $\quad$ **end if**
12: **end if**
13: **Adaptive Sharing**
14: **if** $(t \mod R) = 0$ **then**
15: $\quad$ Share predictions with $\mathcal{P}_k$
16: **else**
17: $\quad z \leftarrow \mathrm{Uniform}([0,1])$
18: $\quad$ **if** $z \leq r_{(k,n)}$ **then**
19: $\quad\quad$ Share predictions with $\mathcal{P}_k$
20: $\quad$ **end if**
21: **end if**
22: **return** $\left(r_{(n,k)}^{t}, \mathcal{L}_{DL_{(n,k)}}\right)$

---

these goals, we need to solve the following two sub-problems: (i) **Reputation Scoring** (RS): How to efficiently evaluate the reliability/reputation $r_{(n,k)}$ of participant $\mathcal{P}_k$ in the context of collaborative learning of model $\mathcal{M}_{\theta_n^*}$ belonging to $\mathcal{P}_n$?, (ii) **Adaptive Sharing** (AS): How to regulate knowledge transfer among participants based on the reputation score to achieve collaborative fairness?.

### 4.1 Reputation Scoring

Several reputation scoring methods have been proposed in the FL literature (Xu & Lyu, 2021; Zhang et al., 2021). These reputation scores are generally used by the FL server either for client selection or for weighted aggregation. In contrast to FL, where a global model is learned by a central server, each participant builds its own local model in PDL. Hence, it is not possible to compute a single reputation score for a PDL participant.

Instead, each participant has its own local estimate of the reputation of other participants. Hence, the same participant may be assigned different reputation scores by their collaborators.

The proposed reputation scoring is based on the intuition that the gradient alignment between local CE loss and pairwise DL is a strong indicator of the utility of $\mathcal{P}_k$ to $\mathcal{P}_n$. For example, if the two gradients are completely misaligned (either because the data distributions of $\mathcal{P}_k$ and $\mathcal{P}_n$ do not have any overlap or because $\mathcal{P}_k$ is malicious), attempting to learn from $\mathcal{P}_k$ is likely to harm the learning of $\mathcal{P}_n$, rather than being beneficial. On the other hand, if there is perfect alignment, the two gradients reinforce each other and accelerate the model learning at $\mathcal{P}_n$. In fact, if KL divergence is used for pairwise DL computation, the gradients based on the local CE loss and pairwise DL will be closely aligned only when the predictions made by $\mathcal{P}_k$ on $\mathcal{P}_n$'s data closely match the ground-truth labels available with $\mathcal{P}_n$. This is because CE loss measures the "distance" between $\sigma_T(\mathcal{M}_{\theta_n}(\boldsymbol{x}))$ and $y$, while DL loss measures the "distance" between $\sigma_T(\mathcal{M}_{\theta_n}(\boldsymbol{x}))$ and $\sigma_T(\mathcal{M}_{\theta_k}(\boldsymbol{x}))$. These

---

**Algorithm 2** Private Decentralized Learning (PDL) based on Proposed CYCle Protocol

---

**Input**: Number of participants $N$, maximum communication rounds $t_{max}$, and learning rate $\eta$
**Assumption**: Each $\mathcal{P}_n$ has $\mathcal{D}_n = \{\boldsymbol{x}_{j,n}, y_{j,n}\}_{j=1}^{|\mathcal{D}_n|}$

1: $\mathcal{M}_{\theta_n^0} \leftarrow$ Local training at $\mathcal{P}_n, \ \forall \ n \in \mathcal{N}$
2: **for** each round $t = 0, 1, \cdots, t_{max}$ **do**
3:      **for** each participant $n \in \mathcal{N}$ **do**
4:          $\mathcal{L}_{CE_n} \leftarrow$ Compute cross-entropy loss of $\mathcal{P}_n$
5:          $\nabla_{\theta_n}\mathcal{L}_{CE_n} \leftarrow$ Compute the gradient of $\mathcal{L}_{CE_n}$
6:          **for** each participant $k \in \mathcal{N}\backslash\{n\}$ **do**
7:             $(r_{(n,k)}^t, \mathcal{L}_{DL_{(n,k)}}) \leftarrow \text{CYCle}(t, k, \nabla\mathcal{L}_{CE_n})$
8:          **end for**
9:          $\lambda_{(n,k)} \leftarrow r_{(n,k)}^t$
10:         $\mathcal{L}_{DL_n} \leftarrow \sum_{k \in \mathcal{N}\backslash\{n\}} \lambda_{(n,k)} \cdot \mathcal{L}_{DL_{(n,k)}}$
11:         $\mathcal{L}_n \leftarrow \mathcal{L}_{CE_n} + \lambda_0 \cdot \mathcal{L}_{DL_n}$
12:         $\theta_n^t \leftarrow \theta_n^{t-1} - \eta\nabla_{\theta_n}\mathcal{L}_n$
13:      **end for**
14: **end for**

---

losses will lead to aligned gradients only when $\sigma_T(\mathcal{M}_{\theta_k}(\boldsymbol{x}))$ is close to $y$. This justifies our choice of using the gradient alignment between CE loss and pairwise DL as the basis for reputation scoring.

The reputation of a participant $\mathcal{P}_k$ from the perspective of $\mathcal{P}_n$ (denoted as $r_{(n,k)}$) is computed in two steps. First, the gradient (mis)alignment metric $s_{(n,k)}$ is computed as:

$$s_{(n,k)} = \frac{1 - \cos\left(\nabla_{\theta_n}\mathcal{L}_{CE_n}, \nabla_{\theta_n}\mathcal{L}_{DL_{(n,k)}}\right)}{2}, \tag{6}$$

where $\nabla_{\theta_n}\mathcal{L}_{CE_n}$ and $\nabla_{\theta_n}\mathcal{L}_{DL_{(n,k)}}$ are the gradients of cross entropy and distillation losses, respectively, with respect to the current parameters $\theta_n$ of the participant $\mathcal{P}_n, n \in \mathcal{N}$, and $k \in \mathcal{N}\backslash\{n\}$. Note that $s_{(n,k)} \to 0$ indicates better gradient alignment. We empirically observed that when the gradient alignment $s_{(n,k)}$ is below a threshold $\tau_{opt}$, collaboration with $\mathcal{P}_k$ is mostly beneficial to $\mathcal{P}_n$. This is because as long as the predictions of $\mathcal{P}_k$ are closer to the ground-truth at $\mathcal{P}_n$, learning from $\mathcal{P}_k$ will help $\mathcal{P}_n$. In contrast, when $s_{(n,k)}$ is above a threshold $\tau_{max}$, collaboration with $\mathcal{P}_k$ becomes harmful to $\mathcal{P}_n$. Based on these observations, we compute the reputation score by applying a soft clipping function to $s_{(n,k)}$. Specifically, the reputation score in the current round $\tilde{r}_{(n,k)} = h(s_{(n,k)})$, where $h$ is defined as:

$$h(s) = \max\left(0, \min\left(1, \frac{s - \tau_{max}}{\tau_{opt} - \tau_{max}}\right)\right). \tag{7}$$

To minimize variations due to stochasticity, the reputation score is updated using a momentum factor, i.e., $r_{(n,k)}^t = \alpha r_{(n,k)}^{(t-1)} + (1-\alpha)\tilde{r}_{(n,k)}$, where $0 < \alpha < 1$ is the momentum hyperparameter and $r_{(n,k)}^t$ is the final reputation score in round $t$ ($t > 0$). When $t = 0$, $r_{(n,k)}^t = \tilde{r}_{(n,k)}$. Henceforth, we drop the index $t$ for convenience. A higher reputation score $r_{(n,k)}$ implies that updates from $\mathcal{P}_k$ are useful for model learning at $\mathcal{P}_n$. Hence, $\mathcal{P}_n$ can directly utilize $r_{(n,k)}$ to weight ($\lambda_{(n,k)}$) the pairwise distillation loss. While it is possible to perform reputation scoring and dynamically update the weights $\lambda_{(n,k)}$ after every collaboration round, it can also be performed periodically (after a fixed number of rounds) to minimize computational costs.

## 4.2 Adaptive Sharing

Once the reputation scores are computed, $\mathcal{P}_n$ can use $r_{(n,k)}$ to choose its collaborators wisely. Specifically, $\mathcal{P}_n$ can decide if it needs "*to share or not to share*" the distillation loss $\mathcal{L}_{DL_{(k,n)}}$ with $\mathcal{P}_k$. To be precise, $\mathcal{P}_n$

computes and sends $\mathcal{L}_{DL_{(k,n)}}$ to $\mathcal{P}_k$ with probability $r_{(n,k)}$. This adaptive sharing mechanism plays a key role in ensuring collaborative fairness. In order to receive updates from $\mathcal{P}_n$ in the current collaboration round, participant $\mathcal{P}_k$ needs to maintain a high reputation score with $\mathcal{P}_n$. This, in turn, requires the sharing of updates that are useful for model learning at $\mathcal{P}_n$ in the previous round.

The above reputation-based adaptive sharing scheme has the following advantages: (1) It incentivizes $\mathcal{P}_k$ to share honest and useful updates with $\mathcal{P}_n$. Sharing noisy or malicious updates will hurt $\mathcal{P}_k$'s reputation score and hence, dampen $\mathcal{P}_k$'s ability to learn from $\mathcal{P}_n$. (2) If $\mathcal{P}_n$ finds that $\mathcal{P}_k$ has a low reputation score, it will send updates less frequently to $\mathcal{P}_k$, thereby saving valuable computational and communication resources. Note that in the PDL framework, privacy-preserving DL loss computation is often computationally expensive. However, there is one limitation in the proposed protocol. If $\mathcal{P}_n$ is malicious, it can deviate from the protocol and stop sending updates to $\mathcal{P}_k$ even though $r_{(n,k)}$ is high. While this can be addressed by penalizing participants (by reducing their reputation score) for not sharing updates, such an approach is not desirable because it will force participants to keep sharing updates even if they are not benefiting from the collaboration. Hence, we go for a compromise solution, where all the parties are forced to share updates after every $R$ collaboration rounds, and the reputation scores are recalibrated based on these responses.

## 4.3 Theoretical Comparison with FedAvg via Mean Estimation

To illustrate the theoretical benefit of CYCle over traditional FL approaches such as FedAvg (McMahan et al., 2017), we consider a simple mean estimation problem with two clients ($N = 2$), each minimizing $f_k(w) = (w - \theta_k)^2$ using limited local samples. Let $\widehat{\theta}_1 \sim \mathcal{N}(\theta_1, \gamma^2)$ and $\widehat{\theta}_2 \sim \mathcal{N}(\theta_2, \gamma^2)$ denote the empirical estimates of each client, where $\gamma^2 = \nu^2/N$ is the estimation variance.

In FedAvg, the clients compute the global model as $w = (\widehat{\theta}_1 + \widehat{\theta}_2)/2$. However, as shown in prior work (Cho et al., 2022), the probability that this shared model outperforms a standalone estimate is exponentially suppressed with increasing data heterogeneity:

$$\mathbb{P}\left(F_k(w) \le F_k(\widehat{\theta}_k)\right) \le 2\exp\left(-\frac{\gamma_G^2}{5\gamma^2}\right), \tag{8}$$

where $\gamma_G^2 = ((\theta_1 - \theta_2)/2)^2$ quantifies heterogeneity. This shows that FedAvg performs poorly when the client distributions are dissimilar.

In contrast, CYCle uses a reputation-weighted aggregation: $w = (1 - r/2) \cdot \widehat{\theta}_1 + r/2 \cdot \widehat{\theta}_2$, where the reputation score $r$ depends on the empirical distance between $\widehat{\theta}_1$ and $\widehat{\theta}_2$.

**Theorem 4.1.** *The probability that the model obtained through CYCle's collaborative aggregation improves upon a standalone model is lower bounded as*

$$\mathbb{P}\left(F_k(w) \le F_k(\widehat{\theta}_k)\right) \ge \frac{1}{8}\exp\left(-\frac{1}{4\gamma^2}\right).$$

Unlike FedAvg, this bound is independent of heterogeneity $\gamma_G^2$, highlighting CYCle's robustness to client diversity. Full derivations and proofs are provided in Appendix A.

## 4.4 Extension to Gossip-SGD

CYCle can be integrated with decentralized optimization algorithms such as Gossip-SGD (Boyd et al., 2006; Koloskova et al., 2019) by modifying the communication mechanism through a dynamic, reputation-driven mixing matrix. In classical Gossip-SGD, each client averages model updates with neighbors using a static mixing matrix. We replace this matrix with one that evolves based on alignment scores between gradient directions, capturing the degree of collaborative benefit.

Formally, during each communication round, each node computes the pairwise cosine similarity with its neighbors' gradients and maps the resulting score using a softmax function controlled by a hyperparameter $\beta$. The updated mixing matrix $W$ is then used to sample a binary interaction mask $S \sim \text{Bernoulli}(W)$, ensuring that communication occurs only with beneficial peers. This mechanism balances collaboration and robustness by probabilistically encouraging high-alignment links while discouraging noisy exchanges.

We detail the full algorithm and its integration into Gossip-SGD in Appendix C, where we also describe the mapping functions and sampling strategy. Empirical results on the Fed-ISIC2019 dataset confirm that this extension maintains fairness and improves utility under realistic, non-i.i.d. data conditions.

## 5    Experiments

**Datasets. CIFAR-10** (Krizhevsky et al., 2009) is a dataset consisting of 60000 images of size $32 \times 32$ pixels, categorized into 10 classes with 6000 images per class. It has 50000 training and 10000 test samples. **CIFAR-100** dataset (Krizhevsky et al., 2009) shares similarities with the CIFAR-10 dataset, but it consists of 100 classes with 600 samples per class. The training and test sets are divided in the same way as in CIFAR-10. The CIFAR-100 dataset is used with data augmentation (random rotation (up to 15 degrees), random crop, and horizontal flip). **Fed-ISIC2019** (Ogier du Terrail et al., 2022) (from FLamby benchmark) is a multi-class dataset of dermoscopy images comprising $23,247$ images with 8 different melanoma classes and high label imbalance (ranging from 49% to less than 1% for class 0 to 7). The dataset is designed for 6 clients based on the centers. Train/test split is: $(9930/2483), (3163/791), (2691/672), (1807/452), (655/164), (351/88)$.

### 5.1    Experimental Setup

**Baseline approaches.** We consider several existing FL algorithms as baselines: vanilla FL based on FedAvg (FedAvg) (McMahan et al., 2017), FL based on cosine gradient Shapley value (CGSV) (Xu et al., 2021), collaborative fairness in FL (CFFL) (Lyu et al., 2020), and robust and fair FL (RFFL) (Xu & Lyu, 2021). The above methods are designed for FL with central orchestration. As mentioned earlier, the baseline VPDL method is based on CaPriDe learning (Tastan & Nandakumar, 2023). We also evaluate standalone (SA) accuracy, where each participant trains its ML model on its local dataset without any collaboration with others. Furthermore, we report the results for these baseline methods on CIFAR-10 and CIFAR-100. We do not compare against Cronus or similar decentralized methods, as they focus on robustness under adversarial conditions and omit fairness evaluation. Since our goal is to promote fair collaboration, we instead benchmark against fairness-aware FL methods. We leave integration with robustness-centric frameworks for future work.

**Participant data splitting strategies.** We implement three types of strategies to split the training dataset among the participants: (i) **homogeneous**, (ii) **heterogeneous**, and (iii) **imbalanced dataset sizes** (denoted as 'imbalanced'). A homogeneous setting refers to the case where each participant gets an equal number of data points per class. In contrast, the heterogeneous method assigns a varying number of data points to each participant, based on a Dirichlet($\delta$) distribution. The parameter $\delta$ reflects the degree of non-IID characteristics within the dataset, where smaller values of $\delta$ lead to a more heterogeneous setting, while larger values tend towards an IID setting. To determine the specific allocation, we sample $p \sim Dir_N(\delta)$ and assign a fraction $p_n$ of the total data samples to $\mathcal{P}_n$. We employ these settings for CIFAR-10 and CIFAR-100 with the following number of participants: $N = 2, 5, 10$. In all our experiments, we utilize all training samples (e.g., all 50000 samples of CIFAR-10 and CIFAR-100). In the case of the imbalanced data distribution, we utilize a custom function that relies on parameters $\kappa$ and $m$, where $\kappa$ determines the proportion of data points received by each of the $m$ chosen participants. The remaining $(N - m)$ participants then share the remaining data among themselves. We explore this setting with $N = 5$ participants and consider the following configurations: (i) imbalanced ($\kappa = 0.8$, $m = 1$), (ii) imbalanced ($\kappa = 0.35$, $m = 2$), and (iii) imbalanced ($\kappa = 0.6$, $m = 1$). For example, in the last case, one participant holds 60% of the data, while the remaining 4 participants get 10% of the data each.

**Evaluation metrics.** We assess the accuracy of the trained models based on their mean validation accuracy (MVA) and mean collaboration gain (MCG) across all participants. To evaluate collaborative fairness, we use the collaboration gain spread (CGS). Reputation scores are reported in a matrix form to determine the relative contribution of each client.

### 5.2    Experimental Results

We start by benchmarking existing algorithms on the CIFAR-10 dataset. Table 1 summarizes the results for different splitting strategies that involve $N = 5$ participants. Our key findings are as follows:

Table 1: Performance comparison on CIFAR-10 dataset: Validation accuracy evaluated with $N = 5$ participants. The top section of the table presents the performance of our proposed framework compared to the FedAvg algorithm. The bottom part of the table compares the collaboration gain and fairness of our proposed CYCle algorithm with existing works (rows 6-9). We use MVA (↑), MCG (↑) and CGS (↓) as eval. metrics.

| Setting | Homogeneous | | | Dirichlet (0.5) | | | Imbalanced (0.8, 1) | | | Imbalanced (0.35, 2) | | | Imbalanced (0.6, 1) | | |
|---|---|---|---|---|---|---|---|---|---|---|---|---|---|---|---|
| Metric | MVA | MCG | CGS | MVA | MCG | CGS | MVA | MCG | CGS | MVA | MCG | CGS | MVA | MCG | CGS |
| FedAvg | 90.60 | 7.20 | 0.48 | 88.76 | 21.40 | 4.31 | 90.17 | 26.35 | 14.51 | 90.34 | 13.12 | 9.61 | 90.16 | 16.28 | 9.11 |
| VPDL | 84.98 | 1.58 | 0.71 | 74.27 | 6.91 | 2.31 | 67.18 | 3.36 | 3.58 | 78.71 | 1.49 | 4.01 | 75.43 | 1.55 | 2.45 |
| CYCle | 86.33 | 2.93 | **0.32** | 76.93 | 9.57 | **2.07** | 69.26 | 5.44 | **2.56** | 81.12 | 3.89 | **2.65** | 76.72 | 2.83 | **1.38** |
| CFFL | 62.65 | 2.21 | 0.87 | 49.04 | 2.49 | 2.35 | 58.66 | 3.48 | 4.39 | 65.08 | 9.99 | 9.95 | 65.20 | 14.35 | 9.79 |
| RFFL | 61.50 | 1.05 | 0.95 | 48.52 | 1.96 | 1.63 | 56.27 | 1.08 | **2.45** | 58.51 | 3.43 | 7.46 | 51.35 | 0.50 | 6.74 |
| CGSV | 63.27 | 2.83 | 0.94 | 54.45 | 7.90 | 3.40 | 55.96 | 0.78 | 9.32 | 61.99 | 6.91 | 11.34 | 61.34 | 10.49 | 11.23 |
| CYCle | 71.95 | 11.51 | **0.58** | 51.21 | 4.65 | **1.11** | 61.80 | 6.62 | 2.79 | 68.26 | 13.17 | **7.12** | 64.93 | 14.08 | **6.22** |

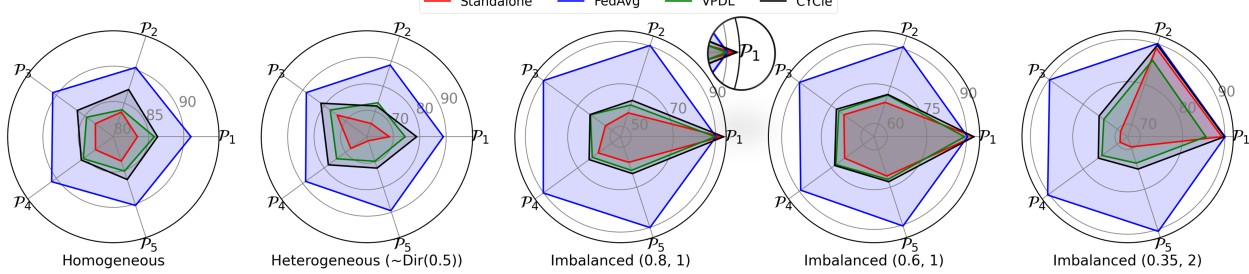

Figure 3: Per-participant performance comparison on CIFAR-10 dataset: Validation accuracy evaluated with $N = 5$ participants.

- We assess the performance of three algorithms – FedAvg, VPDL, and CYCle – using the ResNet18 architecture. These results are shown in Figure 3 and in the upper section of Table 1 (rows 3-5). FedAvg has better MCG compared to other methods due to its centralized FL approach, which involves aggregating and sharing full model parameters, resulting in good overall utility. However, when it comes to collaborative fairness, FedAvg falls short of being fair to participants who contribute a larger amount of data. In particular, the CYCle approach exhibits significantly lower CGS values compared to the FedAvg algorithm in the imbalanced settings. Additionally, our method consistently provides positive collaboration gains to all participants in various settings. This is not the case for FedAvg or VPDL, which is evident in the outcomes for $\mathcal{P}_1$ in the imbalanced $(0.8, 1)$ and $(0.6, 1)$ settings.

- For comparison with other fairness-aware FL algorithms, we utilize a custom CNN architecture used in Xu et al. (2021). This is necessary because CGSV works better on simpler architectures. The results are shown in Fig. 4 and the bottom part of Tab. 1 (rows 6-9). Our method has better or comparable CG values when compared to CFFL, RFFL, and CGSV in most of the scenarios. In terms of fairness, we surpass other approaches by achieving the lowest CGS values in all but one scenario: the imbalanced $(0.8, 1)$ setting, where it is on par with RFFL. Notably, CYCle consistently achieves positive gains for all participants, whereas other methods degrade performance for the most contributing participant (see Figure 4).

Next, we assess the performance of our approach on the CIFAR-10 and CIFAR-100 datasets with different numbers of participants ($N \in \{5, 10\}$) and data splitting scenarios (Table 2). In all settings, we consistently achieve a positive collaboration gain for all participants, and the CYCle method consistently has lower CGS compared to VPDL, demonstrating improved fairness across participants. Furthermore, we observe that when integrated with Gossip-SGD (DSGD), CYCle maintains this trend: while the baseline DSGD achieves high MCG values due to aggressive mixing, it often results in large CGS values, indicating significant disparity

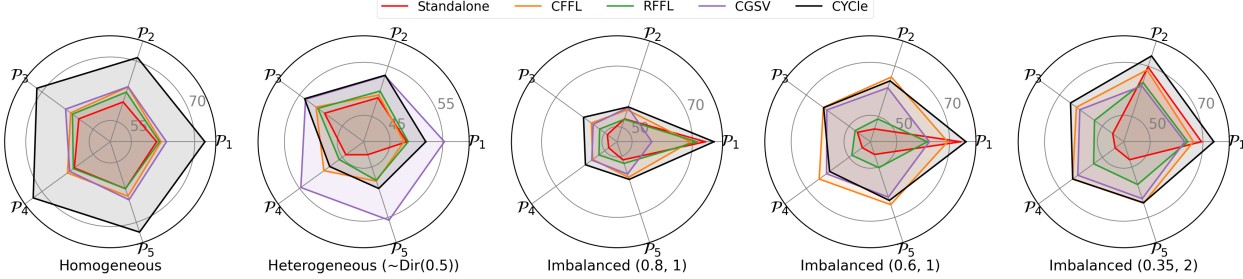

Figure 4: Per-participant performance comparison on CIFAR-10 dataset using custom CNN architecture from (Xu et al., 2021): Validation accuracy evaluated with $N = 5$ participants.

Table 2: Collaboration gain and spread (MCG, CGS) on CIFAR-10 and CIFAR-100 datasets and the given partition strategies for $N = \{5, 10\}$ using our proposed algorithm. DSGD refers to the Gossip-SGD algorithm.

| Dataset | | | CIFAR-10 | | | | | | CIFAR-100 | | | | |
|---|---|---|---|---|---|---|---|---|---|---|---|---|---|
| N | | | 5 | | | 10 | | | 5 | | | 10 | |
| Split | Method | MVA | MCG | CGS | MVA | MCG | CGS | MVA | MCG | CGS | MVA | MCG | CGS |
| Homogeneous | VPDL | 84.98 | 1.58 | 0.71 | 71.01 | 1.92 | 1.58 | 56.74 | 7.58 | 1.20 | 40.87 | 10.42 | 2.48 |
| | CYCle | 86.24 | 2.84 | **0.37** | 73.66 | 4.57 | **0.71** | 57.56 | 8.40 | **0.69** | 41.17 | 10.72 | **1.15** |
| | DSGD | 84.05 | 8.86 | **0.16** | 80.57 | 13.75 | 0.48 | 56.45 | 19.28 | 0.19 | 49.79 | 23.95 | 0.38 |
| | CYCle | 84.22 | 9.03 | **0.16** | 79.84 | 13.02 | **0.30** | 56.07 | 18.90 | **0.12** | 49.41 | 23.57 | **0.23** |
| Dirichlet (0.5) | VPDL | 74.27 | 6.91 | 2.31 | 50.64 | 1.24 | 2.24 | 46.36 | 10.51 | 2.59 | 27.70 | 3.41 | 2.80 |
| | CYCle | 76.93 | 9.57 | **2.13** | 53.82 | 4.42 | **1.98** | 47.98 | 12.13 | **0.68** | 28.47 | 4.18 | **1.07** |
| | DSGD | 81.83 | 22.98 | 3.54 | 77.16 | 28.26 | 4.92 | 50.16 | 21.74 | 2.84 | 44.38 | 24.12 | 2.11 |
| | CYCle | 81.14 | 22.30 | **2.60** | 75.82 | 26.92 | **3.71** | 49.61 | 21.19 | **0.95** | 43.91 | 23.65 | **1.16** |
| Dirichlet (2.0) | VPDL | 81.26 | 3.38 | 3.72 | 69.36 | 5.51 | 5.00 | 56.31 | 12.43 | 2.31 | 39.34 | 9.71 | 3.16 |
| | CYCle | 83.13 | 5.25 | **2.42** | 72.33 | 8.48 | **4.83** | 57.61 | 13.73 | **2.14** | 39.93 | 10.30 | **1.97** |
| | DSGD | 83.66 | 12.64 | 0.97 | 79.86 | 18.25 | 2.29 | 55.53 | 21.19 | 0.97 | 50.21 | 25.89 | 0.61 |
| | CYCle | 83.48 | 12.45 | **0.68** | 79.64 | 18.03 | **1.87** | 56.36 | 22.01 | **0.89** | 50.59 | 26.27 | **0.57** |
| Dirichlet (5.0) | VPDL | 83.32 | 2.39 | 1.94 | 72.37 | 4.54 | 3.96 | 58.18 | 9.80 | 1.38 | 41.86 | 10.03 | 2.92 |
| | CYCle | 84.80 | 3.86 | **1.15** | 74.83 | 7.00 | **3.72** | 59.67 | 11.29 | **1.08** | 41.92 | 10.09 | **1.38** |
| | DSGD | 83.72 | 10.02 | 0.56 | 80.38 | 15.65 | 1.17 | 56.06 | 20.02 | 0.54 | 50.33 | 24.94 | 0.67 |
| | CYCle | 83.85 | 10.15 | **0.33** | 80.46 | 15.73 | **0.77** | 55.87 | 19.83 | **0.48** | 49.62 | 24.24 | **0.43** |

among participants. In contrast, CYCle significantly reduces CGS without compromising MCG, highlighting its ability to regulate knowledge flow more equitably in decentralized and peer-to-peer settings.

**Reputation scoring visualization.** Figure 5 visualizes the reputation scores computed in three collaboration rounds: $t = 0, 25, 75$ using heatmap plots. In Figure 5a, where participants have imbalanced data (with $\mathcal{P}_1$ having 80% of the data), our method can already identify distinct patterns of each participant's data value from the beginning of the collaboration. The heatmap figure emphasizes that higher importance is given to

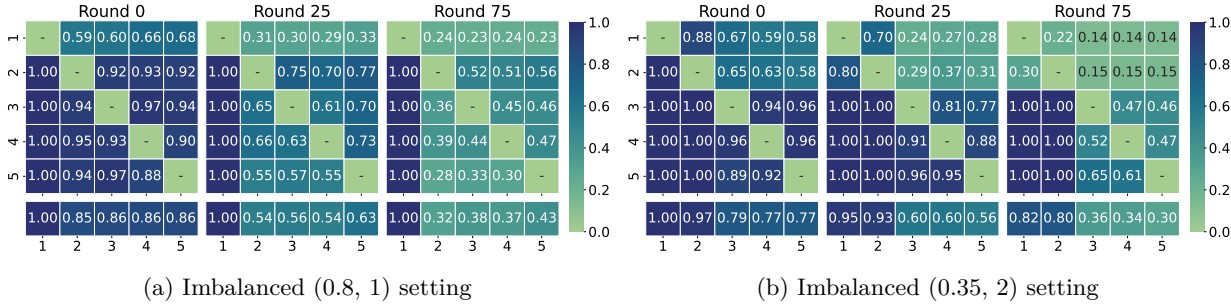

(a) Imbalanced (0.8, 1) setting      (b) Imbalanced (0.35, 2) setting

Figure 5: Heatmap visualization of reputation scores for imbalanced settings when $N = 5$.

$\mathcal{P}_1$ by all the other participants. As the collaboration progresses, the reputation of $\mathcal{P}_1$ intensifies further, while the reputation scores of other participants decrease. Toward the end of the collaboration, the values converge to a state in which participants with an equal amount of data receive equal reputation scores. A similar pattern can be observed in Figure 5b, where the first two participants possess larger amounts of data. While the first five rows of the heatmap figures depict the raw reputation scores calculated using Equation 7, the sixth row at the bottom represents the average reputation scores aggregated over all the rounds.

Table 3: Validation accuracies of participants ($\mathcal{P}_1, ..., \mathcal{P}_5$) with $\mathcal{P}_5$ having varying rates of label flipping. The last column refers to the average accuracy of honest participants ($\mathcal{P}_1, ..., \mathcal{P}_4$). The corresponding figure with reputation scores: Fig. 6.

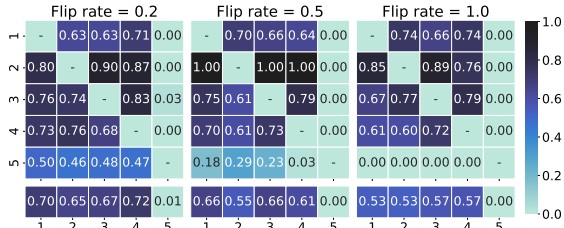

| Setting | $\mathcal{P}_1$ | $\mathcal{P}_2$ | $\mathcal{P}_3$ | $\mathcal{P}_4$ | $\mathcal{P}_5$ | avg |
|---|---|---|---|---|---|---|
| Standalone ($\mathcal{B}_n$) | 83.42 | 83.67 | 83.15 | 83.13 | 83.63 | 83.34 |
| Flip rate = 0.0 | 86.12 | 87.03 | 86.33 | 85.62 | 86.10 | 86.28 |
| Flip rate = 0.2 | 84.95 | 84.5 | 84.5 | 86.45 | 60.95 | 85.10 |
| Flip rate = 0.5 | 84.25 | 84.65 | 84.65 | 83.25 | 55.80 | 84.20 |
| Flip rate = 1.0 | 83.9 | 84.5 | 83.95 | 85.1 | 10.75 | 84.36 |

Figure 6: Heatmap of reputation scores when $\mathcal{P}_5$ undergoes label flipping. Higher flip rates lead to $\mathcal{P}_5$ being rapidly identified as malicious by others.

**Free Rider Study.** In this study involving $N = 5$ participants with a homogeneous data splitting scenario, we simulate varying degrees of label flipping at participant $\mathcal{P}_5$. We tested the label flipping rates of 0.2, 0.5, and 1.0. From Figure 6, it is evident that honest participants (participants $\mathcal{P}_1, ..., \mathcal{P}_4$) quickly discern the anomaly with $\mathcal{P}_5$, detecting it as a potential threat. Consequently, they assign a reputation score of 0.0 to $\mathcal{P}_5$, indicating a cessation of updates to this participant. On the other hand, $\mathcal{P}_5$ accurately recognized that the other participants were beneficial when the label flipping rate was at 0.2, but this positive assessment drops to zero for the case when the label space of $\mathcal{P}_5$ becomes completely different.

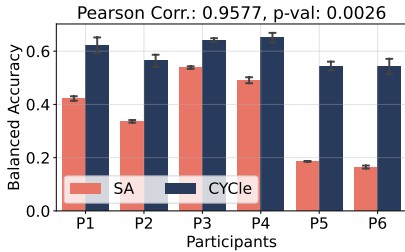

Figure 7: Per-participant accuracy under CYCle (Fed-ISIC2019).

Table 3, aligned with Figure 6, details the values of $\mathcal{B}_n$ and $\mathcal{A}_n$. It reveals that, in the presence of flipped labels, the collaboration gain declines, reflecting a loss of a participant that holds 20% of the data.

**Evaluation on Real-World Medical Dataset (Fed-ISIC2019).** To validate our framework in real-world heterogeneous settings, we evaluate CYCle on the non-IID Fed-ISIC2019 dataset (Ogier du Terrail et al., 2022). This benchmark features complex inter-client shifts in class distributions and image types.

Table 4: CYCle vs. Gossip-SGD on MCG/CGS (Fed-ISIC2019).

| Method | MCG | CGS |
|---|---|---|
| Gossip-SGD (Complete) | 30.26 | 15.64 |
| CYCle | 23.79 | **10.94** |

Figure 7 shows the per-client validation accuracy using our CYCle extension on top of Gossip-SGD. Despite differences in client data sizes and class coverage, CYCle maintains high fairness across participants, as evidenced by a strong correlation between client contribution and reward ($\rho = 0.9577$). Table 4 confirms that CYCle achieves the lowest CGS while preserving collaborative utility.

The complete experimental setup, alternate topologies, and detailed visualizations (including reputation dynamics and data distribution) are provided in Appendix G.3.

## 6    Conclusion

This paper proposed a novel reputation-based adaptive sharing algorithm to promote collaborative fairness, specifically tailored for decentralized learning scenarios. By leveraging gradient alignment, CYCle dynamically adjusts knowledge transfer based on participants' contributions, ensuring positive and equitable collaboration gains. We further extended the protocol to operate seamlessly over gossip-based decentralized learning algorithms. Through theoretical and empirical analyses, this work has illustrated the algorithm's superior capabilities in achieving collaborative fairness.

## Broader Impact Statement

**Privacy Protection Measures.** CYCle inherits the secure knowledge-sharing protocol of the CaPriDe learning framework (Tastan & Nandakumar, 2023), which uses fully homomorphic encryption (FHE) to safeguard sensitive data. Specifically, each client only transmits *encrypted prediction logits* to peers, **never raw data (e.g. medical images from Fed-ISIC2019)** or model parameters. These encrypted outputs are used solely to compute a local distillation loss on peer-held datasets, and they are never decrypted by any party. This ensures that all sensitive information remains protected throughout the training process.

**Abuse Risk and Mitigation.** A common concern in collaborative learning is the potential for *knowledge monopolization*, where dominant participants attempt to suppress or filter contributions from less performant peers. CYCle directly addresses this risk via its similarity-based, bidirectional collaboration mechanism. Specifically, clients benefit most when they are aligned with others (as defined by gradient similarity), which discourages unilateral exclusion. Furthermore, the soft contribution mapping function $h(s)$ makes it suboptimal for any client to reject collaboration entirely, as doing so reduces their own update quality. This design promotes a balanced and inclusive exchange of knowledge.

## Acknowledgments

This material is partly based on work supported by the Office of Naval Research N00014-24-1-2168.

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

# Contents

# A  Mean Estimation under CYCle Protocol

**Standard FL.**  The vanilla FedAvg algorithm, a standard FL approach, does not ensure an optimal final model for all participants in a federated setting. Consider a simple scenario involving two clients, $N = 2$, each aiming to estimate the mean of their data distribution by minimizing the loss function $f_k(w) = (w - \theta_k)^2$. However, these clients can't compute the true mean directly due to having only $N_k$ samples from their distributions, denoted $e_{k,j} \sim \mathcal{N}(\theta_k, \nu^2), \forall j \in [N_k]$. Consequently, they minimize the empirical loss function $F_k(w) = (w - \widehat{\theta}_k)^2 + (\widehat{\theta}_k - \theta_k)^2$, where $\widehat{\theta}_k = \frac{1}{N_k} \sum_{j=1}^{N_k} e_{k,j}$, leading to $\widehat{w}_k = \widehat{\theta}_k$ as the minimizer of $F_k(w)$.

Additionally, the following quantities are defined:

$$\gamma^2 := \frac{\nu^2}{N}; \qquad \gamma_G^2 = \left( \frac{\theta_1 - \theta_2}{2} \right)^2 \tag{9}$$

Note that the distribution of the empirical means themselves is distributed normally, due to the linear additivity property of independent normal random variables. This setting is similar to (Cho et al., 2022; 2024), albeit with a few modifications.

$$\widehat{\theta}_1 \sim \mathcal{N}(\theta_1, \gamma^2); \qquad \widehat{\theta}_2 \sim \mathcal{N}(\theta_2, \gamma^2) \tag{10}$$

**Lemma A.1.** *(Cho et al., 2022) The probability (likelihood) that the model obtained from collaborative training outperforms the standalone training model is upper-bounded by $2 \exp \left( - \frac{\gamma_G^2}{5 \gamma^2} \right)$ in a standard FL framework.*

*Proof.* We analyze the probability that the global model obtained using FedAvg algorithm is superior to one trained independently (standalone). The model in the standard FL approach is defined as:

$$w = \frac{\widehat{\theta}_1 + \widehat{\theta}_2}{2} \tag{11}$$

Then, the **usefulness** of the federated model can be measured in the following way:

$$\mathbb{P}\left( (w - \theta_1)^2 \le (\widehat{\theta}_1 - \theta_1)^2 \right)$$

$$= \mathbb{P}\left( \left( \frac{\widehat{\theta}_1 + \widehat{\theta}_2}{2} - \theta_1 \right)^2 \le (\widehat{\theta}_1 - \theta_1)^2 \right) \tag{12}$$

$$= \mathbb{P}\left( \left( \frac{\widehat{\theta}_1 + \widehat{\theta}_2}{2} - \theta_1 \right)^2 - (\widehat{\theta}_1 - \theta_1)^2 \le 0 \right) \tag{13}$$

$$= \mathbb{P}\left( \left( \frac{\widehat{\theta}_2 - \widehat{\theta}_1}{2} \right)^2 + 2(\widehat{\theta}_1 - \theta_1)\left( \frac{\widehat{\theta}_2 - \widehat{\theta}_1}{2} \right) \le 0 \right) \tag{14}$$

$$= \mathbb{P}\left( \left\{ \left( \frac{\widehat{\theta}_2 - \widehat{\theta}_1}{2} \right)^2 + 2(\widehat{\theta}_1 - \theta_1)\left( \frac{\widehat{\theta}_2 - \widehat{\theta}_1}{2} \right) \le 0 \right\} \cap \left\{ \widehat{\theta}_2 > \widehat{\theta}_1 \right\} \right)$$

$$+ \mathbb{P}\left( \left\{ \left( \frac{\widehat{\theta}_2 - \widehat{\theta}_1}{2} \right)^2 + 2(\widehat{\theta}_1 - \theta_1)\left( \frac{\widehat{\theta}_2 - \widehat{\theta}_1}{2} \right) \le 0 \right\} \cap \left\{ \widehat{\theta}_2 \le \widehat{\theta}_1 \right\} \right) \tag{15}$$

$$= \mathbb{P}\left( \left\{ \left( \frac{\widehat{\theta}_2 - \widehat{\theta}_1}{2} \right) + 2(\widehat{\theta}_1 - \theta_1) \le 0 \right\} \cap \left\{ \widehat{\theta}_2 > \widehat{\theta}_1 \right\} \right)$$

$$+ \mathbb{P}\left( \left\{ \left( \frac{\widehat{\theta}_2 - \widehat{\theta}_1}{2} \right)^2 + 2(\widehat{\theta}_1 - \theta_1)\left( \frac{\widehat{\theta}_2 - \widehat{\theta}_1}{2} \right) \le 0 \right\} \cap \left\{ \widehat{\theta}_2 \le \widehat{\theta}_1 \right\} \right) \tag{16}$$

$$\leq \mathbb{P}\left(\left(\frac{\widehat{\theta}_2 - \widehat{\theta}_1}{2}\right) + 2(\widehat{\theta}_1 - \theta_1) \leq 0\right) + \mathbb{P}\left(\widehat{\theta}_2 - \widehat{\theta}_1 \leq 0\right) \tag{17}$$

$$= \mathbb{P}(Z_1 \leq 0) + \mathbb{P}(Z_2 \leq 0) \qquad \text{where } Z_1 \sim \mathcal{N}\left(\gamma_G, \frac{5}{2}\gamma^2\right), Z_2 \sim \mathcal{N}(2\gamma_G, 2\gamma^2) \tag{18}$$

$$\leq \exp\left(-\frac{\gamma_G^2}{5\gamma^2}\right) + \exp\left(-\frac{\gamma_G^2}{\gamma^2}\right) \leq 2\exp\left(-\frac{\gamma_G^2}{5\gamma^2}\right) \tag{19}$$

where (15) uses $\mathbb{P}(A) = \mathbb{P}(A \cap B) + \mathbb{P}(A \cap B^C)$, (17) uses $\mathbb{P}(A \cap B) \leq \mathbb{P}(A)$, (18) uses the linear additivity property of independent normal random variables and (10), (19) uses a Chernoff bound.

$\square$

**CYCle Algorithm.** The CYCle algorithm uniquely incorporates the reputations of each client during the aggregation of their computed means. Considering the scenario from the perspective of Client 1, who possesses an estimate $\widehat{\theta}_1$ and receives $\widehat{\theta}_2$ from Client 2, the aggregation is expressed as:

$$w_1 = \left(1 - \frac{r_{1,2}}{2}\right)\widehat{\theta}_1 + \frac{r_{1,2}}{2}\widehat{\theta}_2 \tag{20}$$

As CYCle operates on a decentralized principle, each client develops its final model based on individual reputations denoted by $r$. For example, for Client 2, the model is:

$$w_2 = \left(1 - \frac{r_{2,1}}{2}\right)\widehat{\theta}_2 + \frac{r_{2,1}}{2}\widehat{\theta}_1 \tag{21}$$

Notably, $w_1$ and $w_2$ are not necessarily the same due to the distinct reputation scores. The following discussion will simplify the notation by dropping indices and focusing on the aggregation process from the perspective of Client 1.

**Reputation Scoring.** In the CYCle algorithm, each participant calculates the reputation of their collaborators by assessing the distance between their shared means, denoted as $d = \left(\frac{\widehat{\theta}_2 - \widehat{\theta}_1}{2}\right)^2$. This distance is then used to determine their reputations as follows:

$$r = \begin{cases} 1.0 & \text{if } d \leq 1 \\ 2 - d & \text{if } 1 < d \leq 2 \\ 0.0 & \text{if } d > 2 \end{cases} \tag{22}$$

Here, $d$ represents the empirical heterogeneity parameter, referred to as $\widehat{\gamma}_G^2$. The rationale for the assigned values is based on the degree of heterogeneity (22):

- If $\widehat{\gamma}_G^2 < 1$, it indicates minimal heterogeneity, suggesting that participants can benefit significantly from collaboration.

- If $\widehat{\gamma}_G^2 > 2$, it signals substantial heterogeneity, advising participants to avoid collaboration.

Thus, the reputation scores are directly influenced by the empirical heterogeneity observed between the participants.

**Cost Function.** To evaluate the CYCle algorithm against the standard, centralized FL algorithm, we consider the empirical losses each participant ends up solving. In FedAvg, all clients achieve the common mean:

$$w = \frac{\widehat{\theta}_1 + \widehat{\theta}_2}{2} \tag{23}$$

and the empirical loss for Client 1 is:

$$F_1(w) = F_1\left(\frac{\widehat{\theta}_1 + \widehat{\theta}_2}{2}\right) = \widehat{\gamma}_G^2 + (\widehat{\theta}_1 - \theta_1)^2 \tag{24}$$

This formulation shows that each client encounters non-reducible loss terms in the FedAvg algorithm. For the CYCle algorithm, the model aggregation is reputation-weighted:

$$w = \left(1 - \frac{r}{2}\right)\widehat{\theta}_1 + \frac{r}{2}\widehat{\theta}_2 \tag{25}$$

yielding the empirical loss:

$$F_1(w) = r^2\widehat{\gamma}_G^2 + (\widehat{\theta}_1 - \theta_1)^2 \tag{26}$$

In CYCle, the empirical heterogeneity parameter $\widehat{\gamma}_G^2$ is adjusted using the reputation scores $r$, allowing each client to potentially achieve a final model that is as good as or better than their standalone model. Conversely, in FedAvg, all clients share the same final model. As the heterogeneity between clients increases $(\gamma_G^2)$, the efficacy (usefulness) of the global model significantly diminishes, as highlighted in Lemma A.1. This comparison illustrates the advantage of CYCle in managing heterogeneity among the participants to enhance individual outcomes.

**Minimizing the CYCle Objective: Ensuring Positive Outcomes.** Unlike standard FL where increasing true heterogeneity $\gamma_G^2$ drastically reduces the usefulness of the global model (as noted in Lemma A.1), CYCle modulates the aggregation phase using reputation scores which are derived from the empirical heterogeneity parameter $\widehat{\gamma}_G^2$ (22).

**Theorem A.1** (Restated Theorem 4.1). *The probability (likelihood) that the model obtained through collaboration (w) surpasses a standalone-trained model $(\widehat{\theta})$ is lower bounded by $\frac{1}{8}\exp\left(-\frac{1}{4\gamma^2}\right)$ in CYCle algorithm.*

*Proof.* We assess the performance (usefulness) of the collaborative model for client $i$ using:

$$\mathbb{P}\left((w_i - \theta_i)^2 \leq (\widehat{\theta}_i - \theta_i)^2\right) \tag{27}$$

This analysis includes both of the scenarios: (i) $\widehat{\theta}_1 = \widehat{\theta}_2$ which directly implies a probability of 1.0, and (ii) a significant difference between $\widehat{\theta}_1$ and $\widehat{\theta}_2$ ($\widehat{\gamma}_G^2 \gg 0$), leading to no collaboration, also resulting in a probability of 1.0. The CYCle algorithm is not interested in obtaining a final global model and each client is self-interested in minimizing their own objective functions.

**Case 1:** $\widehat{\gamma}_G^2 \leq 1$

$$\mathbb{P}\left((w_1 - \theta_1)^2 \leq (\widehat{\theta}_1 - \theta_1)^2\right) \tag{28}$$

$$= \mathbb{P}\left((w_1 - \widehat{\theta}_1)^2 + 2(w_1 - \widehat{\theta}_1)(\widehat{\theta}_1 - \theta_1) \leq 0\right) \tag{29}$$

$$= \mathbb{P}\left(\left(\frac{\widehat{\theta}_2 - \widehat{\theta}_1}{2}\right)^2 + 2\left(\frac{\widehat{\theta}_2 - \widehat{\theta}_1}{2}\right)(\widehat{\theta}_1 - \theta_1) \leq 0\right) \tag{30}$$

$$= \mathbb{P}\left(\left\{\left(\frac{\widehat{\theta}_2 - \widehat{\theta}_1}{2}\right)^2 + 2\left(\frac{\widehat{\theta}_2 - \widehat{\theta}_1}{2}\right)(\widehat{\theta}_1 - \theta_1) \leq 0\right\} \cap \{\widehat{\theta}_2 > \widehat{\theta}_1\}\right)$$

$$+ \mathbb{P}\left(\left\{\left(\frac{\widehat{\theta}_2 - \widehat{\theta}_1}{2}\right)^2 + 2\left(\frac{\widehat{\theta}_2 - \widehat{\theta}_1}{2}\right)(\widehat{\theta}_1 - \theta_1) \leq 0\right\} \cap \{\widehat{\theta}_2 \leq \widehat{\theta}_1\}\right) \tag{31}$$

$$\geq \mathbb{P}\left(\{1 + 2(\widehat{\theta}_1 - \theta_1) \leq 0\} \cap \{\widehat{\theta}_2 > \widehat{\theta}_1\}\right) \tag{32}$$

$$= \mathbb{P}\left(\left\{\widehat{\theta}_1 - \theta_1 \leq -\frac{1}{2}\right\} \cap \{\widehat{\theta}_2 > \widehat{\theta}_1\}\right) \tag{33}$$

$$= \mathbb{P}(\widehat{\theta}_1 < \widehat{\theta}_2)\, \mathbb{P}\big(\widehat{\theta}_1 - \theta_1 \le -\frac{1}{2} | \widehat{\theta}_1 < \widehat{\theta}_2\big) \tag{34}$$

$$\ge \mathbb{P}(\widehat{\theta}_1 < \widehat{\theta}_2)\, \mathbb{P}\big(\widehat{\theta}_1 - \theta_1 \le -\frac{1}{2}\big) \tag{35}$$

$$= \mathbb{P}(\widehat{\theta}_1 < \widehat{\theta}_2)\, \mathbb{P}\big(Z > \frac{1}{2\gamma}\big) \quad \text{where } Z \sim \mathcal{N}(0,1) \tag{36}$$

$$\ge \frac{1}{8} \exp\big(-\frac{1}{4\gamma^2}\big) \tag{37}$$

where (30) uses the reputation score $r = 1$ when $\widehat{\gamma}_G^2 < 1$ (refer to (22)), (31) uses $\mathbb{P}(A) = \mathbb{P}(A \cap B) + \mathbb{P}(A \cap B^C)$, (32) uses the definition of $\widehat{\gamma}_G^2 \le 1$ and the second part of the expression can be dropped since $\widehat{\gamma}_G$ can take values from $-1$ to $0$, it can be assigned zero probability, (35) uses $\mathbb{P}(A|B) \ge \mathbb{P}(A)$, (36) uses $\widehat{\theta}_1 - \theta_1 \sim \mathcal{N}(0, \gamma^2)$, (37) $\mathbb{P}(\widehat{\theta}_1 < \widehat{\theta}_2) \ge \frac{1}{2}$ and $\mathbb{P}(Z \ge x) \ge \frac{1}{4} \exp(-x^2)$ where $Z \sim \mathcal{N}(0,1)$. The similar bound can be found for the second client.

**Case 2:** $1 < \widehat{\gamma}_G^2 \le 2$

In this case, we can follow the similar approach above and we get:

$$\mathbb{P}\big((w_1 - \theta_1)^2 \le (\widehat{\theta}_1 - \theta_1)^2\big) \tag{38}$$

$$= \mathbb{P}\big((w_1 - \widehat{\theta}_1)^2 + 2(w_1 - \widehat{\theta}_1)(\widehat{\theta}_1 - \theta_1) \le 0\big) \tag{39}$$

$$= \mathbb{P}\big(\big(\frac{r}{2}(\widehat{\theta}_2 - \widehat{\theta}_1)\big)^2 + r(\widehat{\theta}_2 - \widehat{\theta}_1)(\widehat{\theta}_1 - \theta_1) \le 0\big) \tag{40}$$

$$= \mathbb{P}\big(r\big(\frac{\widehat{\theta}_2 - \widehat{\theta}_1}{2}\big)^2 + 2\big(\frac{\widehat{\theta}_2 - \widehat{\theta}_1}{2}\big)(\widehat{\theta}_1 - \theta_1) \le 0\big) \tag{41}$$

$$\ge \mathbb{P}\big(\{1 + 2(\widehat{\theta}_1 - \theta_1) \le 0\} \cap \{\widehat{\theta}_2 > \widehat{\theta}_1\}\big) \tag{42}$$

$$\ge \frac{1}{8} \exp\big(-\frac{1}{4\gamma^2}\big) \tag{43}$$

The reason behind getting the same bound as in Case 1 is that the expression in (41) is maximized when $r = (2 - \widehat{\gamma}_G^2) = 1$, which necessitates $\widehat{\gamma}_G \approx 1$, when $\widehat{\theta}_1 < \widehat{\theta}_2$.

**Case 3:** $\widehat{\gamma}_G^2 > 2$

$$\mathbb{P}\big((w_1 - \theta_1)^2 \le (\widehat{\theta}_1 - \theta_1)^2\big) = \big((\widehat{\theta}_1 - \theta_1)^2 \le (\widehat{\theta}_1 - \theta_1)^2\big) = 1 \tag{44}$$

where $w_1 = \big(1 - \frac{r}{2}\big)\widehat{\theta}_1 + \frac{r}{2}\widehat{\theta}_2 = \widehat{\theta}_1$, since $r = 0$ for $\widehat{\gamma}_G^2 > 2$ (Eqs. 25, 22).

As such, CYCle effectively adapts to various degrees of heterogeneity, ensuring that collaboration invariably enhances or matches the standalone training performance. This is quantitatively supported by a lower bound of $\frac{1}{8} \exp\big(-\frac{1}{4\gamma^2}\big)$, demonstrating robustness against heterogeneity effects.

$\square$

**Scope and Limitations.** The result in Theorem 4.1 is derived under a simplified two-client mean estimation setting. While this toy model is useful to illustrate how collaboration may fail under heterogeneity and how CYCle can mitigate that through similarity-based reputation, it is not intended as a universal lower bound for all learning scenarios. In particular, the lower bound on success probability does not generalize trivially to more complex settings (e.g., neural networks, high-dimensional tasks, or $N > 2$ clients).

Extending the analysis to larger populations introduces new technical challenges. Specifically, when $N > 2$, (i) the reputation becomes a full matrix rather than a scalar, (ii) true-mean gaps form a vector rather than a single $\gamma_G$, and (iii) the same noisy estimate $\widehat{\theta}_i$ appears in multiple clients' aggregates, introducing

cross-correlations. The bias term then depends on the worst-case alignment across all pairs, while the variance term includes complex cross-terms from covariance expansion. Bounding these quantities is non-trivial and remains an open problem.

We include this example primarily to explain the mechanism by which data heterogeneity undermines fairness in standard FL approaches, and to motivate our design choices in CYCle. Developing more general guarantees is an exciting direction for future work.

## B    Numerical Experiments

### B.1    Varying Degrees of Heterogeneity

For the first experiment, we adopt the scenario outlined in previous studies (Cho et al., 2022; 2024). For the simulation involving two clients, we define the true means of the clients as $\theta_1 = 0, \theta_2 = \gamma_G$ where $\gamma_G \in [0, 5]$. The empirical means, $\widehat{\theta}_1$ and $\widehat{\theta}_2$, are sampled from the distribution $\mathcal{N}(\theta_1, 1)$ and $\mathcal{N}(\theta_2, 1)$ respectively, with equal sample sizes ($N_1 = N_2 = N$). The average usefulness is calculated over 10000 runs.

The results, depicted in Figure 8, highlight the differing impacts of heterogeneity on the final model's usefulness. Under the FedAvg protocol, as heterogeneity increases, the global model's usefulness drops drastically, potentially disadvantaging each client by diverging from their optimal solutions (empirical means). Conversely, the CYCle approach exhibits an increase in the usefulness of the resultant models ($w_i$) irrespective of the heterogeneity level. This indicates an effective mechanism within CYCle for clients to discern and select good collaborators based on the proximity of their updates. In scenarios of pronounced heterogeneity, resulting in low reputation scores, clients are incentivized to rely more heavily on their own updates, effectively mitigating collaboration when it proves detrimental.

### B.2    Imbalanced Data

In this experiment, we consider a scenario where both clients share the same true mean $\theta_1 = \theta_2 = 0$, effectively setting the true heterogeneity parameter $\gamma_G^2 = 0$. Despite this, the clients differ in their number of

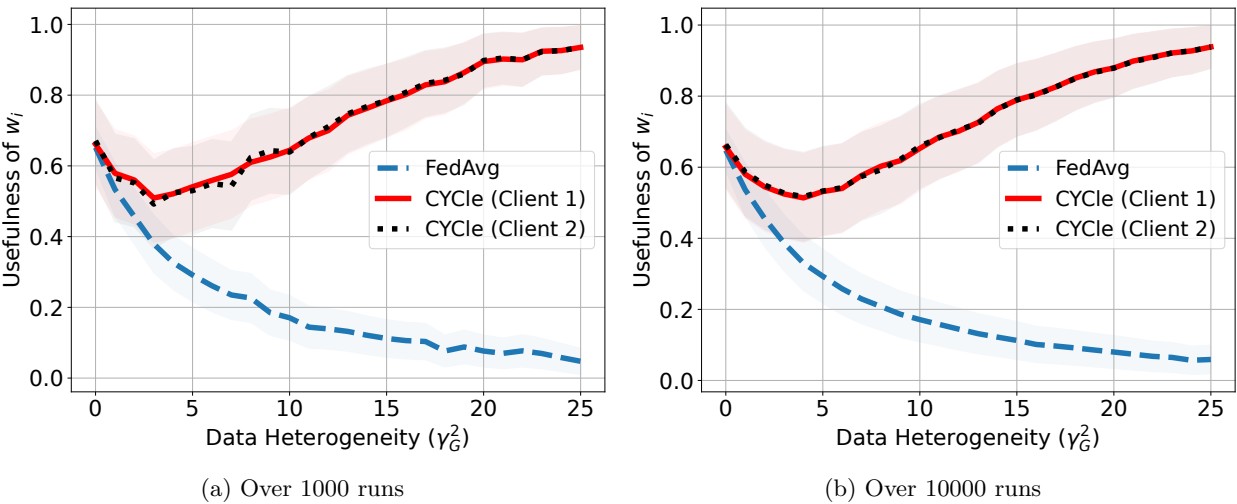

(a) Over 1000 runs                                (b) Over 10000 runs

Figure 8: Numerical study on the usefulness of the model obtained after collaboration ($\mathbb{P}((w_i - \theta_i)^2 \leq (\widehat{\theta}_i - \theta_i)^2)$) in two client mean estimation. In FedAvg, a single global model is used, whereas in CYCle, each client computes their model ($w_i$) independently.

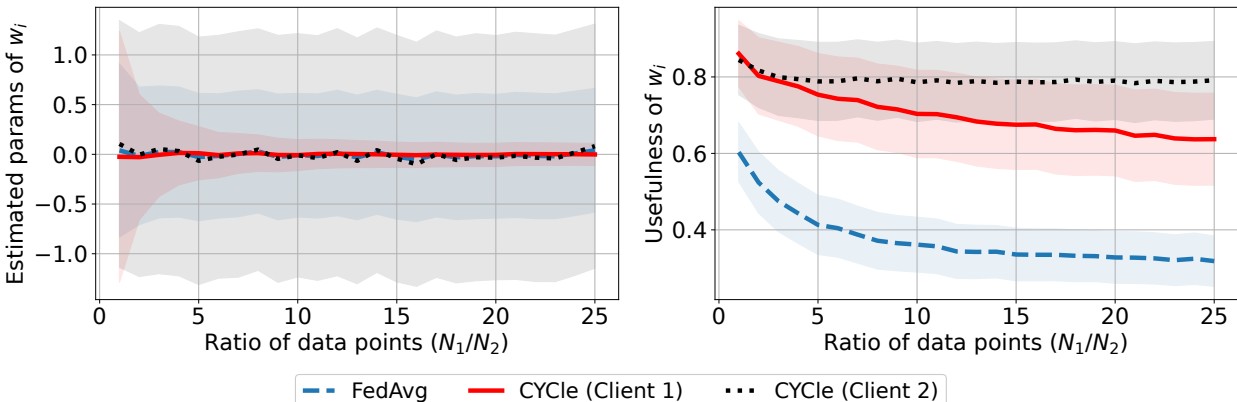

Figure 9: Numerical study on the impact of data imbalance on model usefulness $(\mathbb{P}((w_i - \theta_i)^2 \leq (\widehat{\theta}_i - \theta_i)^2))$ and the depiction of $\theta_i, \sigma_i$ estimates for $w_i \sim \mathcal{N}(\theta_i, \sigma_i^2)$ in two client mean estimation across varying degrees of data ratios.

samples, influencing their empirical means. Specifically, $\widehat{\theta}_1$ and $\widehat{\theta}_2$ are drawn from distributions $\mathcal{N}(0, \frac{\sigma^2}{N_1})$ and $\mathcal{N}(0, \frac{\sigma^2}{N_2})$, respectively, with $\sigma^2$ set to 5.

The results, presented in Figure 9, illustrate the differential impacts of collaboration based on sample size. Collaboration tends to benefit Client 2 more than Client 1, with Client 1 being the higher-contributing client in terms of data. The variance of Client 1 decreases as their data ratio increases, suggesting that Client 1 would benefit from limiting trust in Client 2's contributions under significant data imbalances. In contrast, the variance for Client 2 remains relatively the same regardless of the data ratio.

Figure 9 (right side) demonstrates the varying utilities of collaboration for each client. For Client 1, the usefulness of collaboration decreases as the data ratio becomes more skewed, whereas it remains constant for Client 2, implying that the collaboration is useful for Client 2. Notably, the FedAvg approach generally results in worse outcomes compared to both clients in the CYCle scenario, especially as imbalances in data contribution increase.

## C   Extension to Gossip-SGD

This appendix provides detailed background on decentralized optimization and describes how the proposed CYCle protocol is integrated with Gossip-SGD. We include a full derivation of the update rules, communication graph structures, and Algorithm 3. This section supports the main discussion in Section 4.4.

Unlike the distillation-based approach described in Equations 1 and 2, which focuses on federated distillation settings, decentralized algorithms typically aim to solve the *average consensus problem*. This problem is formalized as:

$$\overline{\boldsymbol{x}} := \frac{1}{N} \sum_{i=1}^{N} \boldsymbol{x}_i, \tag{45}$$

where $\boldsymbol{x}_i \in \mathbb{R}^d$ are local vectors distributed across $N$ nodes.

### C.1   Gossip-SGD

**Description of Gossip-SGD.**   Classic decentralized algorithms for solving the average consensus problem are often based on *gossip-based* algorithms (Xiao et al., 2005; Boyd et al., 2006; Koloskova et al., 2019; 2020; Ying et al., 2021) that generate sequences $\{\boldsymbol{x}_i^{(t)}\}_{t \geq 0}$ on every node $i \in [N]$ through iterative updates of the

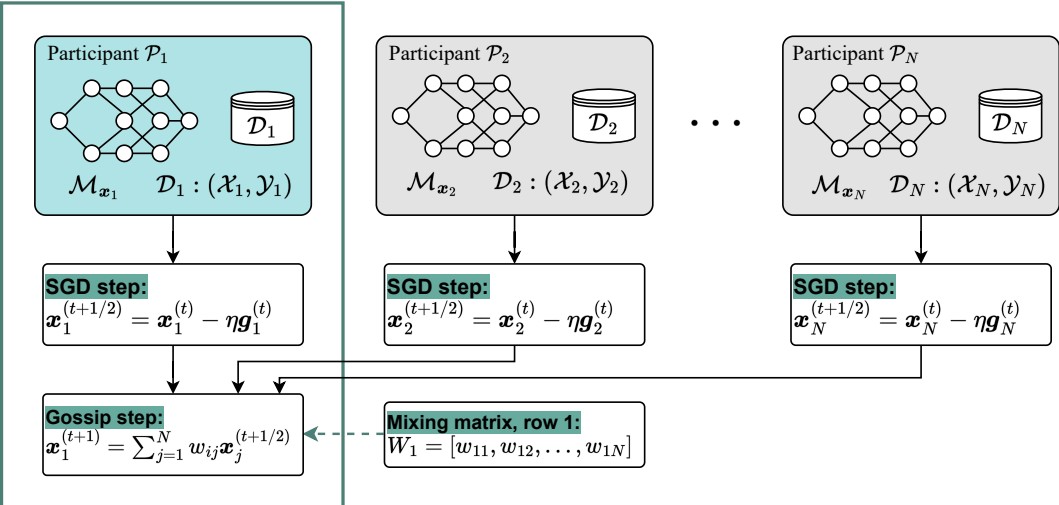

Figure 10: Illustration of the Gossip-SGD algorithm for $N$ participants, focusing on the learning process of participant $\mathcal{P}_1$. As depicted in the figure, each participant performs SGD updates locally, followed by a gossip step where they share and exchange with other participants.

form:

$$\boldsymbol{x}_i^{(t+1)} := \boldsymbol{x}_i^{(t)} + \eta \sum_{j=1}^{N} w_{ij} \Delta_{ij}^{(t)}, \tag{46}$$

where $\eta \in (0, 1]$ is a stepsize, $w_{ij} \in [0, 1]$ are the weights, and $\Delta_{ij}^{(t)} \in \mathbb{R}^d$ is the vector (e.g. gradients) sent from node $j$ to node $i$ at iteration $t$.

Communication occurs only if $w_{ij} > 0$, and no communication is needed when $w_{ij} = 0$. If the weights are symmetric ($w_{ij} = w_{ji}$), they define a communication graph $G = ([N], E)$ with edges $\{i, j\} \in E$ if $w_{ij} > 0$ and self-loops $\{i\} \in E$ for $i \in [N]$. The matrix $W \in \mathbb{R}^{N \times N}$, known as the *mixing* matrix or *gossip* matrix, specifies the connectivity of the network, with $(W)_{ij} = w_{ij}$.

The objective in decentralized optimization is to minimize the global function:

$$\min_{\boldsymbol{x} \in \mathbb{R}^{N \times d}} F(\boldsymbol{x}), \quad \text{where} \quad F(\boldsymbol{x}) = \sum_{i=1}^{N} f_i(\boldsymbol{x}_i), \tag{47}$$

where $f_i$ represents the local objective at node $i$.

To solve the optimization problem in Equation 47, the Gossip-SGD algorithm alternates between the following two steps:

$$\boldsymbol{x}_i^{(t+1/2)} = \boldsymbol{x}_i^{(t)} - \eta \boldsymbol{g}_i^{(t)} \left( \mathbb{E}\left[\boldsymbol{g}_i^{(t)}\right] = \nabla f_i\left(\boldsymbol{x}_i^{(t)}\right) \right) \qquad \triangleright \text{ SGD step} \tag{48}$$

$$\boldsymbol{x}_i^{(t+1)} = \sum_{j=1}^{N} w_{ij} \boldsymbol{x}_j^{(t+1/2)} \qquad \triangleright \text{ Gossip step} \tag{49}$$

Combining both steps, the Gossip-SGD update in matrix form becomes:

$$\boldsymbol{x}^{(t+1)} = W\left(\boldsymbol{x}^{(t)} - \eta \boldsymbol{g}^{(t)}\right), \quad \text{where} \quad \boldsymbol{g}^{(t)} = \left(\boldsymbol{g}_1^{(t)}, \boldsymbol{g}_2^{(t)}, \ldots, \boldsymbol{g}_N^{(t)}\right)^T \in \mathbb{R}^{N \times d}. \tag{50}$$

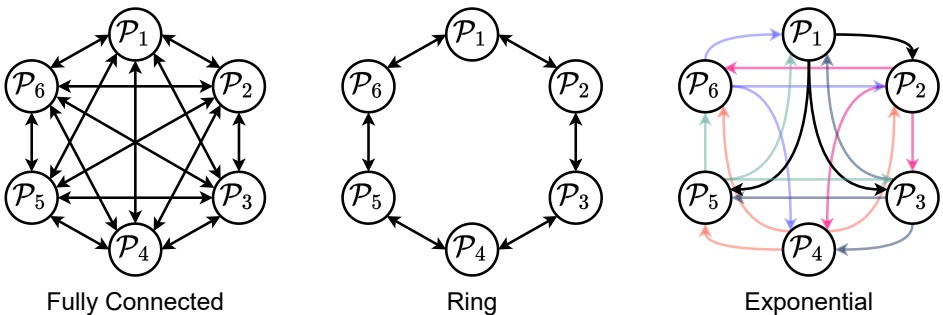

Figure 11: Different types of communication graphs/topologies: fully connected graph, ring graph and exponential graph.

**Communication topologies.** Communication graphs play a critical role in decentralized learning, where these graphs define the communication topology, specifying which nodes (clients) can exchange model updates during training. Unlike standard federated learning (FL), where updates are aggregated by a central server, decentralized FL relies on peer-to-peer exchanges guided by the graph structure. Common topologies, as depicted in Figure 11, include

- *complete graphs*, where all nodes communicate directly;

- *ring graphs*, where nodes only communicate with their immediate neighbors; and

- *exponential graphs*, which optimize connectivity while minimizing communication overhead.

The mixing matrix $W$ is constructed using uniform weights based on the graph topology:

$$[W]_{ij} = \begin{cases} \dfrac{1}{|\mathcal{N}(i)| + 1}, & \text{if } (i, j) \in E \quad \text{or} \quad i = j, \\ 0, & \text{otherwise.} \end{cases} \tag{51}$$

where $|\mathcal{N}(i)|$ denotes the number of neighbors of node $i$.

In our work, we initialize the communication graph as a fully connected graph but incorporate two key mechanisms to make the topology dynamic: (i) reputation scoring and (ii) adaptive sharing (detailed in Section 4). These mechanisms allow the communication graph to evolve during training, enabling improved fairness. We detail the specifics in Section C.2.

## C.2 Integrating CYCle with Gossip-SGD

The algorithm 3 illustrates how we extend the Gossip-SGD algorithm within the framework of our proposed CYCle protocol. The central modification involves the use of a dynamic mixing matrix, which is updated every $R$ communication rounds. This update is guided by a reputation scoring mechanism based on gradient alignment and an adaptive sharing strategy. In this version of CYCle, we adopt a different mapping function than Equation 7, emphasizing the flexibility to use any mapping function, provided it preserves the probabilistic nature (i.e., maps values to the range $[0, 1]$). Specifically, we use a softmax function with a hyperparameter $\beta$. The parameter $\beta$ controls the trade-off between utility and fairness: higher values of $\beta$ penalize smaller cosine similarities, thereby promoting fairness, while lower values prioritize equity in updates, effectively emulating a fully-connected graph topology as a special case. During each communication round, we stochastically sample a binary matrix $S$ from the mixing matrix $W$ using Bernoulli sampling (or a coin toss). The resulting binary mask $S$ determines the communication partners for that round, where updates are exchanged only between nodes corresponding to the entries of 1.0 in $S$. This adaptive sharing mechanism enhances efficiency by facilitating productive collaborations and minimizing unproductive exchanges. Unlike PDL-based methods, which conduct multiple communications within a single FL round and sample updates for distillation, our approach in Gossip-SGD performs the sampling over the communication rounds $T$. We

---

**Algorithm 3** Gossip-SGD algorithm based on CYCle Protocol

---

**Input**: for each node $i \in [N]$ initialize $\boldsymbol{x}_i^{(0)} \in \mathbb{R}^d$, number of communication rounds $T$, stepsize $\eta$, and mixing matrix $W^{(0)}$

1: **for** round $t = 0, 1, \cdots, T$ **do**
2:      **for** each participant $i \in [N]$ in parallel **do**
3:          Compute $\boldsymbol{g}_i^{(t)} \leftarrow \nabla f_i\left(\boldsymbol{x}_i^{(t)}\right)$
4:          **for** each neighbor $j : \{i, j\} \in E$ and $(t \mod R = 0)$ **do**
5:              $s_{ij} \leftarrow \left(1 + \cos\left(\boldsymbol{g}_i^{(t)}, \boldsymbol{g}_j^{(t)}\right)\right)/2$
6:              $r_{ij} \leftarrow \text{arbitrary mapping function}(s_{ij})$          $\triangleright$ softmax function
7:              $w_{ij}^{(t)} \leftarrow \alpha w_{ij} + (1 - \alpha)r_{ij}$, with $\alpha = 0.5$
8:          **end for**
9:          $\boldsymbol{x}_i^{(t+1/2)} = \boldsymbol{x}_i^{(t)} - \eta\boldsymbol{g}_i^{(t)}$          $\triangleright$ stochastic gradient updates
10:      **end for**
11:      Construct $W$, $(W)_{ij} \leftarrow w_{ij}$
12:      Generate stochastic binary mask $S \leftarrow \text{Bernoulli}(W)$
13:      Construct $\widetilde{W} \leftarrow W \odot S^T$ and normalize each row to sum up to 1
14:      $\boldsymbol{x}^{(t+1)} = \widetilde{W}\boldsymbol{x}^{t+1/2}$,    where $\boldsymbol{x}^{t+1/2} = \left(\boldsymbol{x}_1^{t+1/2}, \ldots, \boldsymbol{x}_N^{t+1/2}\right)$          $\triangleright$ gossip averaging
15: **end for**

---

evaluated this approach in Section G.3 on the Fed-ISIC2019 dataset. This dataset, being naturally partitioned and inherently non-IID, serves as a challenging and realistic benchmark for assessing federated learning algorithms.

# D Notation Summary

To improve clarity and readability throughout the paper, we provide Table 5 as a reference for the main notation. This table summarizes the symbols introduced in Section 2.1 and used consistently throughout the paper. By consolidating key mathematical symbols and definitions in one place, the table is intended to serve as a quick reference resource for readers, especially in later sections where several variables reappear in extended formulations. This addition also responds to reviewer feedback requesting improved accessibility of the notation throughout the manuscript.

Table 5: Summary of key notation used throughout the paper.

| Symbol | Description |
|---|---|
| *Participants and Datasets* | |
| $N$ | Total number of participants/clients |
| $\mathcal{N} = \{1, 2, \ldots, N\}$ | Index set of participants |
| $\mathcal{P}_n$ | Participant/client indexed by $n \in \mathcal{N}$ |
| $\mathcal{D}_n$ | Local training dataset of participant $n$ |
| $\tilde{\mathcal{D}}_n$ | Local validation (hold-out) dataset of participant $n$ |
| *Input and Output Spaces* | |
| $\boldsymbol{x} \in \mathcal{X}$ | Input sample from input space $\mathcal{X} \subseteq \mathbb{R}^d$ |
| $y \in \mathcal{Y}$ | Ground-truth label; $\mathcal{Y} = \{1, 2, \ldots, M\}$ with $M$ classes |
| $d$ | Dimensionality of input features |
| $M$ | Number of output classes |
| $\mathcal{Z} \subseteq \mathbb{R}^M$ | Logits space (output of $\mathcal{M}_\theta$) |

Table 5: continued from previous page

| Symbol | Description |
|---|---|
| *Models and Predictions* | |
| $\tilde{\mathcal{M}}_\theta : \mathcal{X} \to \mathcal{Y}$ | Supervised classification model with parameters $\theta$ |
| $\mathcal{M}_\theta : \mathcal{X} \to \mathcal{Z}$ | Logits-generating model mapping inputs to $\mathcal{Z} \subseteq \mathbb{R}^M$ |
| $\theta_n$ | Parameters of participant $n$'s model $\mathcal{M}_{\theta_n}$ |
| $\boldsymbol{z}$ | Logit vector output by $\mathcal{M}_\theta(\boldsymbol{x})$ |
| $\sigma_T$ | Temperature-scaled softmax function: $\sigma_T(\boldsymbol{z}) = \text{softmax}(\boldsymbol{z}/T)$ |
| $\boldsymbol{p}$ | Predicted class probabilities: $\boldsymbol{p} = \sigma_T(\boldsymbol{z})$ |
| $T$ | Temperature parameter controlling softmax sharpness |
| *Losses* | |
| $\mathcal{L}_{CE_n}$ | Cross-entropy loss of participant $n$ on $\mathcal{D}_n$ |
| $\mathcal{L}_{DL_{(n,k)}}$ | Pairwise distillation loss from $k$ to $n$ |
| $\mathcal{L}_{DL_n}$ | Total distillation loss for participant $n$ |
| $\mathcal{L}_n$ | Total training loss of participant $n$ |
| $\lambda_n$ | Weight on distillation loss in $\mathcal{L}_n$ |
| $\lambda_{(n,k)}$ | Weight assigned to collaborator $k$'s contribution in DL for $n$ |
| $\text{KL}(\cdot\|\cdot)$ | Kullback-Leibler divergence used for distillation |
| *Evaluation Metrics* | |
| $\mathcal{B}_n$ | Standalone accuracy of $\mathcal{M}_{\theta_n}$ on $\tilde{\mathcal{D}}_n$ |
| $\mathcal{A}_n$ | Post-collaboration accuracy of $\mathcal{M}_{\theta_n^\star}$ on $\tilde{\mathcal{D}}_n$ |
| $\mathcal{G}_n = \mathcal{A}_n - \mathcal{B}_n$ | Collaborative gain (CG) of participant $n$ |

# E  Computational and Communication Complexity

This section expands on the computational and communication implications of the proposed CYCle protocol, particularly in comparison to baseline methods such as CaPriDe learning (Tastan & Nandakumar, 2023) and Gossip-SGD (Boyd et al., 2006). We quantify overhead associated with encrypted operations, gradient alignment, and communication costs introduced by our protocol, and clarify where CYCle improves upon or matches the efficiency of existing techniques.

## E.1  Computational Complexity

The most significant computational component introduced in CYCle[PDL] stems from its reliance on encrypted distillation loss computations, inherited directly from CaPriDe learning (Tastan & Nandakumar, 2023). As detailed in Table 2 of their work, the per-sample inference time under FHE-based encryption on a commodity CPU is approximately 110 seconds, which represents the cryptographic bottleneck of such secure collaborative schemes. Though recent advancements in the cryptographic field have led to improvements by using GPU-accelerated CKKS libraries, reducing the timing by more than 10 times (Kim et al., 2024; Yang et al., 2024; Jin et al., 2024).

However, unlike CaPriDe learning, where encrypted predictions are computed and transmitted at every training round regardless of peer behavior, CYCle strategically reduces this cost by avoiding encrypted inference in rounds where peer reputation does not warrant a response or when they do not get sampled. This modification reduces the frequency of FHE operations without compromising convergence, leading to substantially lower per-round cryptographic overhead. In practice, this enables CYCle to remain computationally viable at scale, especially in large or heterogeneous client networks.

Another computational addition in CYCle is the use of gradient alignment to guide selective sharing. This is implemented via cosine similarity computations between local gradients obtained with respect to cross-entropy

loss and received peer gradients obtained with respect to distillation losses, which scale linearly with model size. These operations are performed once every $R$ rounds (where $R = 5$ in experiments), and in aggregate, introduce less than a 5% overhead per local step. The periodic nature of these operations, coupled with modest cost, ensures they do not become a computational bottleneck. Similarly, sharing decisions are computed locally and stochastically, requiring no inter-client coordination or additional encryption.

Thus, compared to CaPriDe learning, CYCle significantly reduces encrypted inference load, and compared to Gossip-SGD, it introduces only marginal overhead through lightweight, periodic gradient comparisons.

### E.2 Communication Complexity

In terms of communication, CYCle adheres closely to the decentralized communication patterns of CaPriDe learning and Gossip-SGD but incorporates reputation-aware mechanisms that reduce message exchanges without altering the underlying protocol structure.

**CYCle[PDL].** Similar to CaPriDe learning, CYCle requires clients to share encrypted predictions (logits) with selected clients for participant distillation. However, unlike CaPriDe learning where every client responds to every query, CYCle introduces a reputation-aware mechanism to selectively respond, thus reducing the number of encrypted transmissions. This strategy directly reduces the volume of encrypted data exchanged per round. Empirically, we observe a $35 - 50\%$ reduction in encrypted communication relative to CaPriDe learning across our experimental settings. Each encrypted prediction, typically represented as an FHE ciphertext, incurs a payload size of $1-2$KB, depending on the number of classes $M$, since the pairwise distillation loss is an $M$-dimensional vector.

**CYCle[Gossip-SGD].** When integrated with Gossip-SGD, CYCle does not incur any additional communication overhead beyond what is already required by the baseline. Clients continue to exchange model updates every round, as in standard Gossip protocols. Crucially, the reputation-weighted mixing matrix is never transmitted; each client computes it locally based on observed peer behavior. Additionally, clients may selectively omit update transmissions to low-contributing clients, thereby reducing communication frequency in practice. This selective sharing leads to lower average communication volume per round, especially in heterogeneous or adversarial environments where client contribution quality varies significantly.

## F Experimental Setup. Implementation Details

We keep the same training hyperparameters for all experiments. Cross-entropy loss is used for local model training in all methods. The PDL algorithms (VPDL and CYCle) use Kullback-Leibler (KL) divergence based on predictions as the distillation loss. The optimizer is stochastic gradient descent (SGD) with momentum. The learning rate is initially set to 0.1 and is updated every 25 rounds using a scheduler with a learning rate decay of 0.1. For FL algorithms, we set the number of collaborating rounds to $t_{max} = 100$ and the local update epoch to 1. For the decentralized algorithms, we use 25 epochs of local training before collaboration, followed by 75 rounds of collaboration. The batch size is set to 128 for CIFAR-10 and CIFAR-100, and 32 for the Fed-ISIC2019 dataset. Furthermore, $\alpha = 0.5$, $\lambda_0 = 50$, $\tau_{opt} = 0.25$, and $\tau_{max} = 0.75$. In the experiments involving Gossip-SGD, we set $\beta = 15$ and $R = 5$. We conduct our experiments on NVIDIA A100-PCIE-40GB GPUs on an internal cluster server, with each run utilizing a single GPU. The execution time for each run averages around 1.35 hours.

# G    Additional Experimental Results

This section is organized in the following manner:

1. Heatmap visualizations of reputation scores for the remaining data splitting scenarios, which include homogeneous, Dirichlet (0.5), and imbalanced (0.6, 1) cases.

2. Remaining results of Table 2.

3. Details of the Fed-ISIC2019 data distribution and experimental results of CYCle when combined with Gossip-SGD.

4. Experimental results on the CIFAR-100 dataset.

5. More analysis of the imbalanced data splitting scenario.

6. More analysis of the free rider scenario.

7. Scalability to larger client populations.

8. Trade-off between MCG and CGS.

9. Hyperparameter sensitivity.

## G.1    Reputation Score Visualization

In Figure 12, we present the reputation scores calculated on the CIFAR-10 dataset for a group of five participants ($N = 5$). The figure illustrates the results at rounds $t = 0$, 25, 75, under different data partitioning schemes: (i) homogeneous, (ii) heterogeneous (Dirichlet ($\delta = 0.5$)), (iii) imbalanced ($\kappa = 0.6, m = 1$). In the homogeneous setting, participants have similar reputation scores that correlate with the sizes of the datasets they hold. Over time, these scores trend downwards as the cross-entropy loss gains prominence over the total distillation loss (Eq. 1). The heterogeneous and imbalanced (0.6, 1) scenarios exhibit a similar pattern, where the reputation scores align with the size of the data they possess. In the latter scenario, since $\mathcal{P}_1$ holds

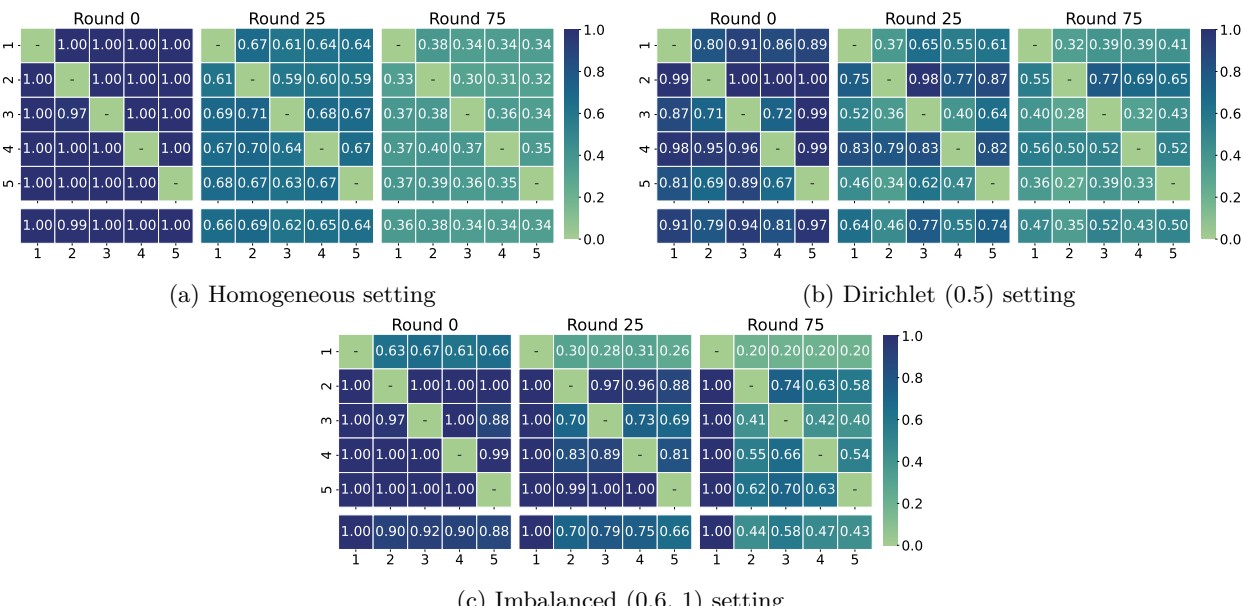

(a) Homogeneous setting          (b) Dirichlet (0.5) setting

(c) Imbalanced (0.6, 1) setting

Figure 12: Heatmap visualization of reputation scores for the given number of participants $N = 5$ and data partition schemes: (a) Homogeneous, (b) Dirichlet (0.5) and (c) Imbalanced (0.6, 1).

60% of the data, its reputation remains consistent, prompting other participants to engage more actively in collaboration to receive updates from this major contributor.

It must be noted that even when dataset sizes are identical among participants, as seen in homogeneous ($\mathcal{P}_1$ to $\mathcal{P}_5$) and imbalanced (0.6, 1) ($\mathcal{P}_2$ to $\mathcal{P}_5$) data splitting scenarios, there may be stochastic variations in data quality resulting in higher reputation scores for some participants compared to others. For instance, in Fig. 12c, $\mathcal{P}_3$ is assigned a marginally higher reputation score by the other participants, despite having an identical dataset size as $\mathcal{P}_2$, $\mathcal{P}_4$, and $\mathcal{P}_5$.

### G.2 Remaining Results of Table 2

This section extends our analysis of collaboration gain and spread (MCG, CGS) for the CIFAR-10 and CIFAR-100 datasets using our proposed algorithm. While Table 2 focused on scenarios with $N \in \{5, 10\}$ participants, the additional results presented in Table 6 cover the scenario with $N = 2$ participants. This study provides further insight into the performance dynamics under a smaller number of participants and complements the study shown in Table 2.

Table 6: Collaboration gain and spread (MCG, CGS) on CIFAR-10 and CIFAR-100 datasets and the given partition strategies for $N = 2$ using our proposed algorithm. This table supplements Table 2, which covers $N = \{5, 10\}$.

| Dataset | | CIFAR-10 | | | CIFAR-100 | | | Dataset | CIFAR-10 | | | CIFAR-100 | | |
|---|---|---|---|---|---|---|---|---|---|---|---|---|---|---|
| Split | Method | MVA | MCG | CGS | MVA | MCG | CGS | Method | MVA | MCG | CGS | MVA | MCG | CGS |
| Homogeneous | VPDL | 91.99 | 0.41 | 0.08 | 71.52 | 4.87 | 0.55 | DSGD | 86.26 | 3.55 | 0.27 | 59.52 | 7.59 | 0.16 |
| | CYCle | 92.58 | 1.00 | **0.01** | 72.03 | 5.38 | **0.22** | CYCle | 86.36 | 3.64 | **0.21** | 59.53 | 7.61 | **0.12** |
| Dirichlet (0.5) | VPDL | 84.90 | 1.45 | 1.88 | 64.99 | 8.95 | 0.31 | DSGD | 84.03 | 16.92 | 12.51 | 55.95 | 10.73 | 8.13 |
| | CYCle | 88.82 | 5.37 | **0.76** | 66.37 | 10.33 | **0.25** | CYCle | 81.63 | 14.52 | **9.24** | 52.63 | 7.42 | **5.11** |
| Dirichlet (2.0) | VPDL | 89.93 | 0.09 | 0.22 | 70.57 | 5.52 | 0.66 | DSGD | 85.89 | 5.10 | **1.51** | 57.91 | 13.59 | **0.71** |
| | CYCle | 91.00 | 1.16 | **0.15** | 72.09 | 7.04 | **0.18** | CYCle | 86.53 | 5.73 | **0.99** | 57.81 | 13.50 | **0.52** |
| Dirichlet (5.0) | VPDL | 90.36 | 0.24 | 0.08 | 71.45 | 4.95 | 0.89 | DSGD | 85.92 | 3.77 | 0.91 | 58.93 | 9.83 | 0.33 |
| | CYCle | 91.29 | 1.17 | **0.03** | 72.12 | 5.62 | **0.36** | CYCle | 86.15 | 4.00 | **0.58** | 58.82 | 9.72 | **0.11** |

### G.3 Fed-ISIC2019 experiments

**Fed-ISIC2019.** We evaluate our framework on the real-world non-IID Fed-ISIC2019 dataset. Note that for easy visualization, the participants in this experiment are sorted in the decreasing order of their dataset size. Figure 14 illustrates the probability that each participant shared updates with their neighbors over $T = 100$ communication rounds. The reported values represent averages; for example, a value of 0.51 indicates that $\mathcal{P}_1$ shared updates with $\mathcal{P}_2$ during 51 out of 100 communication rounds. We observe that our framework captures distribution shift, as evidenced by the fact that the data held by the first client does not overlap with that of other clients (as shown in the UMAP visualization in Figure 1f of Ogier du Terrail et al. (2022)).

**Data Distribution.** Figure 13 displays a heatmap plot illustrating the distribution of data by class among participants. In Figure 14, it is evident that despite $\mathcal{P}_1$ having the bulk of the data, it has a noticeable distribution shift compared to other participants. This indicates that other participants are not allocated certain classes due to the specific distribution settings employed in (Ogier du Terrail et al., 2022). It is noteworthy that initially, $\mathcal{P}_1$ assigns a lower reputation score to $\mathcal{P}_4$. However, over time,

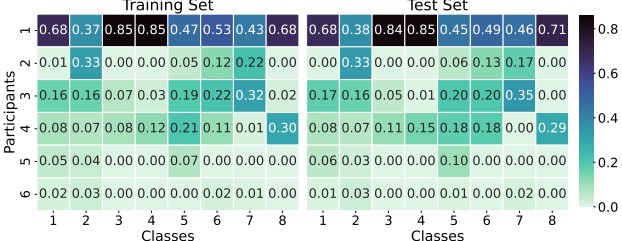

Figure 13: Fed-ISIC2019 data distribution.

$\mathcal{P}_4$'s reputation improves, a change attributable to the exclusive knowledge of class 8 shared only by $\mathcal{P}_1$ and $\mathcal{P}_4$.

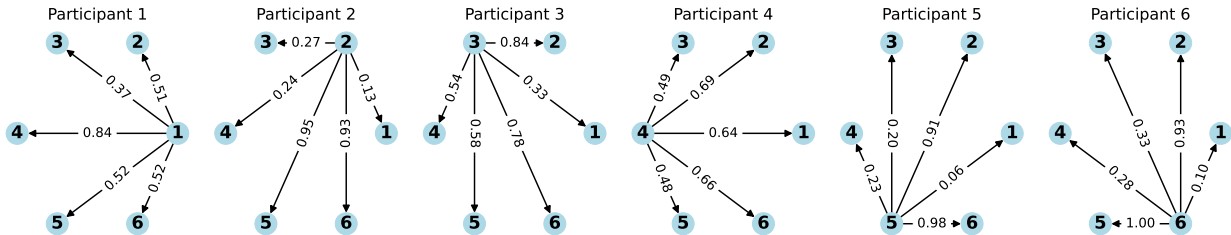

Figure 14: Illustration of communication graph over $T$ communication rounds, representing the percentage of times each participant shared updates with its neighbors.

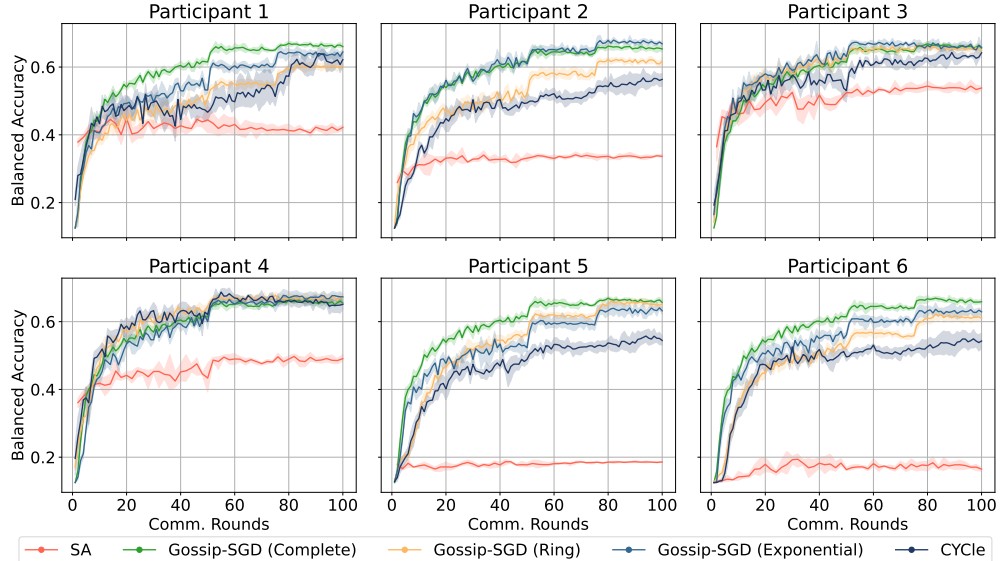

Figure 15: Per-participant performance plots on the Fed-ISIC2019 dataset comparing different methods: standalone training, Gossip-SGD with various topologies (fully connected, ring, exponential), and the proposed CYCle protocol. SA stands for standalone training.

**Experimental Results.** Figure 15 illustrates the performance of the following methods: (i) standalone training, (ii) Gossip-SGD with a complete graph topology, (iii) Gossip-SGD with a ring topology, (iv) Gossip-SGD with an exponential graph topology, and (v) our proposed CYCle algorithm integrated with Gossip-SGD. The experiments span 100 communication rounds, and we report balanced accuracy as the primary metric due to the highly imbalanced nature of the classes. The results are derived from a global test set created by pooling the test sets of all participants to ensure a fair comparison.

As shown in the plots, participants 5 and 6 struggle significantly, achieving performance below 20%. Interestingly, participants 3 and 4 achieve the best performance despite having smaller datasets compared to participants 1 and 2. Among the methods, Gossip-SGD with a complete graph topology achieves the highest overall performance. However, in terms of fairness (as shown in Figure 16a), it fails to provide equitable outcomes (Pearson's corr. coefficient of $-0.2649$), as all participants converge to a uniform final performance regardless of their contributions.

Table 7: MCG and CGS values (Figure 15).

| Method | MCG | CGS |
|---|---|---|
| Gossip-SGD (Complete) | 30.26 | 15.64 |
| Gossip-SGD (Ring) | 27.72 | 14.80 |
| Gossip-SGD (Exponential) | 29.45 | 14.24 |
| CYCle | 23.79 | **10.94** |

In contrast, our CYCle algorithm mitigates this issue by suppressing the collaboration gains of participants contributing less or providing low-quality updates. This results in a high correlation coefficient of 0.9577,

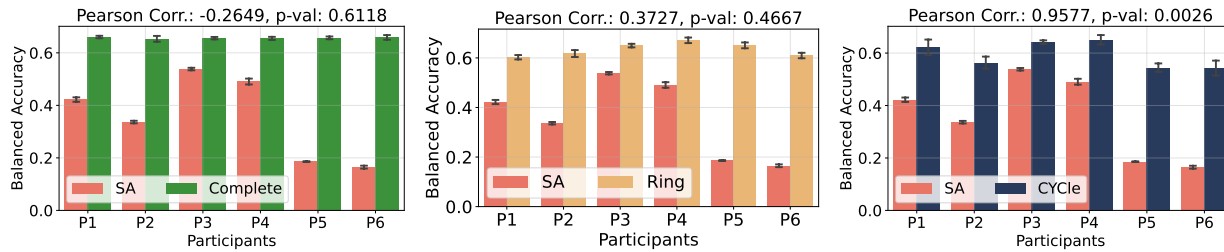

(a) Gossip-SGD with Complete Graph.    (b) Gossip-SGD with Ring Graph.    (c) CYCle with Dynamic Topology.

Figure 16: Per-participant accuracy and Pearson's correlation coefficient of the Gossip-SGD algorithm with fully connected and ring topologies, alongside the performance of our proposed CYCle algorithm. SA stands for standalone training.

with a small p-value of 0.0026, successfully rejecting the null hypothesis (that there is no correlation) at a significance level of 0.05 (see Figure 16c). These results highlight the efficacy of our approach in balancing both performance and fairness across participants. We also report the same plot for Gossip-SGD with a ring topology in Figure 16b, which fails to deliver fair outcomes for all participants. For the results on the MCG and CGS metrics, we refer the reader to Table 7.

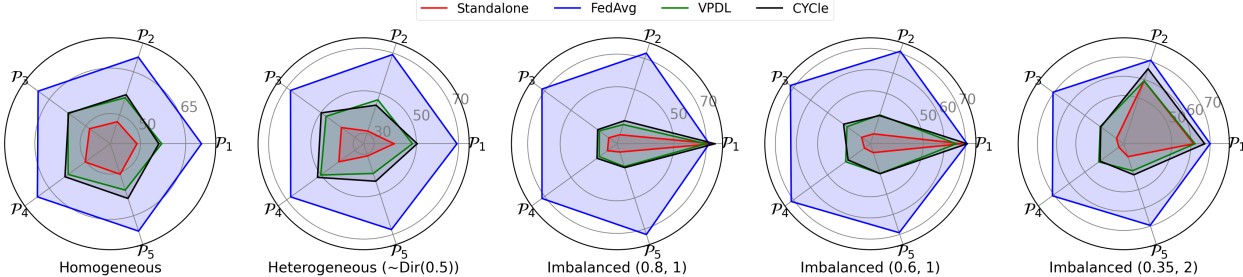

Figure 17: Per-participant performance comparison on CIFAR-100 dataset: Validation accuracy evaluated with $N = 5$ participants.

Table 8: Per-participant performance comparison on CIFAR-100 dataset: Validation accuracy evaluated with $N = 5$ participants.

| Partition | No. | SA | FedAvg | VPDL | CYCle | Partition | No. | SA | FedAvg | VPDL | CYCle |
|---|---|---|---|---|---|---|---|---|---|---|---|
| | 1 | 48.85 | 70.25 | 57.05 | 56.10 | | 1 | 74.50 | 71.80 | 70.05 | 75.30 |
| | 2 | 47.65 | 70.10 | 56.00 | 57.05 | | 2 | 20.80 | 73.45 | 27.40 | 29.85 |
| Homogeneous | 3 | 48.45 | 69.55 | 57.30 | 57.10 | Imbalanced (0.8, 1) | 3 | 21.25 | 72.10 | 28.40 | 29.70 |
| | 4 | 50.25 | 69.85 | 57.25 | 58.55 | | 4 | 22.25 | 72.10 | 28.50 | 30.20 |
| | 5 | 50.60 | 70.35 | 56.10 | 59.00 | | 5 | 20.35 | 73.55 | 29.70 | 30.40 |
| | 1 | 39.40 | 69.20 | 48.15 | 50.35 | | 1 | 61.45 | 70.00 | 62.10 | 67.15 |
| | 2 | 31.30 | 69.25 | 46.80 | 44.15 | | 2 | 59.30 | 70.60 | 59.35 | 65.95 |
| Dirichlet (0.5) | 3 | 38.00 | 67.80 | 47.00 | 49.80 | Imbalanced (0.35, 2) | 3 | 29.65 | 70.55 | 40.15 | 39.80 |
| | 4 | 39.50 | 67.70 | 50.10 | 52.00 | | 4 | 28.10 | 70.95 | 41.05 | 40.30 |
| | 5 | 31.05 | 67.60 | 39.75 | 43.60 | | 5 | 32.05 | 69.55 | 39.75 | 41.95 |

| Partition | No. | SA | FedAvg | VPDL | CYCle |
|---|---|---|---|---|---|
| | 1 | 68.25 | 70.80 | 65.80 | 70.80 |
| | 2 | 29.90 | 70.90 | 39.30 | 39.00 |
| Imbalanced (0.6, 1) | 3 | 30.15 | 71.80 | 38.70 | 40.70 |
| | 4 | 28.50 | 71.35 | 39.45 | 38.05 |
| | 5 | 29.40 | 69.15 | 39.80 | 39.85 |

### G.4 Experiments on CIFAR-100

We perform the same experiment outlined in Table 1 and Figure 3 on the CIFAR-100 dataset. Our goal is to assess the effectiveness and fairness of CYCle under increased label granularity and distributional heterogeneity. Figure 17 and Table 8 depict the performance of each participant within the $N = 5$ group across the various data partitioning strategies detailed in Section 5.1. Similar to the observations in Figure 3, in the imbalanced $(0.8, 1)$ scenario, the FedAvg algorithm results in a decline in accuracy for $\mathcal{P}_1$, indicative of a negative collaboration gain. Following this, we compile the mean validation accuracy (MVA), mean collaboration gain (MCG), and collaboration gain spread (CGS) for each method and data splitting scenario in Table 9.

As in CIFAR-10 experiments, we observe that FedAvg often achieves the highest mean validation accuracy (MVA), particularly in homogeneous and moderately in imbalanced settings. However, this comes at a significant fairness cost. VPDL, while privacy-preserving and decentralized, underperforms in both MVA and MCG across all scenarios. Although it avoids extreme gain inequality (moderate CGS), it fails to leverage meaningful collaboration in data-heterogeneous environments, largely due to its uniform treatment of client contributions.

Table 9: Performance comparison on CIFAR-100 dataset: Validation accuracy evaluated with $N = 5$ participants. The table presents the performance of our proposed framework compared to the FedAvg algorithm. We employ MVA (↑), MCG (↑) and CGS (↓) as evaluation metrics.

| Setting | Homogeneous | | | Dirichlet (0.5) | | | Imbalanced (0.8, 1) | | | Imbalanced (0.35, 2) | | | Imbalanced (0.6, 1) | | |
|---|---|---|---|---|---|---|---|---|---|---|---|---|---|---|---|
| Metric | MVA | MCG | CGS | MVA | MCG | CGS | MVA | MCG | CGS | MVA | MCG | CGS | MVA | MCG | CGS |
| FedAvg | 70.02 | 20.86 | 1.07 | 68.31 | 32.46 | 3.98 | 72.60 | 40.77 | 21.77 | 70.33 | 28.22 | 15.06 | 70.80 | 33.56 | 15.54 |
| VPDL | 56.74 | 7.58 | 1.20 | 46.36 | 10.51 | 2.59 | 36.81 | 4.98 | 4.84 | 48.48 | 6.37 | 5.19 | 44.61 | 7.37 | 4.98 |
| CYCle | 57.56 | 8.40 | **0.69** | 47.98 | 12.13 | **0.68** | 39.09 | 7.26 | **3.30** | 51.03 | 8.92 | **2.40** | 45.68 | 8.44 | **3.00** |

### G.5 Imbalanced Data Study

In this study, we explore a range of values for the parameters $\kappa$ and $m$ in imbalanced settings, involving five participants ($N = 5$), as detailed in Section 5.1. The purpose of this variation is to demonstrate the

Table 10: Performance comparison on CIFAR-10 dataset under imbalanced data: Validation accuracy evaluated with $N = 5$ participants. We employ MVA (↑), MCG (↑) and CGS (↓) as evaluation metrics.

| Setting | Imbalanced (0.4, 1) | | | Imbalanced (0.12, 1) | | | Imbalanced (0.41, 2) | | | Imbalanced (0.23, 2) | | | Imbalanced (0.14, 2) | | |
|---|---|---|---|---|---|---|---|---|---|---|---|---|---|---|---|
| Metric | MVA | MCG | CGS | MVA | MCG | CGS | MVA | MCG | CGS | MVA | MCG | CGS | MVA | MCG | CGS |
| FedAvg | 91.04 | 10.38 | 4.93 | 90.98 | 8.02 | 4.00 | 90.17 | 17.48 | 13.99 | 91.23 | 7.73 | 1.96 | 90.89 | 8.42 | 4.94 |
| VPDL | 83.36 | 2.69 | 3.04 | 86.18 | 3.22 | 2.74 | 74.90 | 2.22 | 4.17 | 85.82 | 2.33 | 2.09 | 86.18 | 3.70 | 3.67 |
| CYCle | 85.02 | 4.35 | **2.26** | 86.78 | 3.82 | **2.41** | 76.71 | 4.02 | **2.27** | 86.48 | 2.98 | **1.17** | 86.92 | 4.44 | **3.47** |

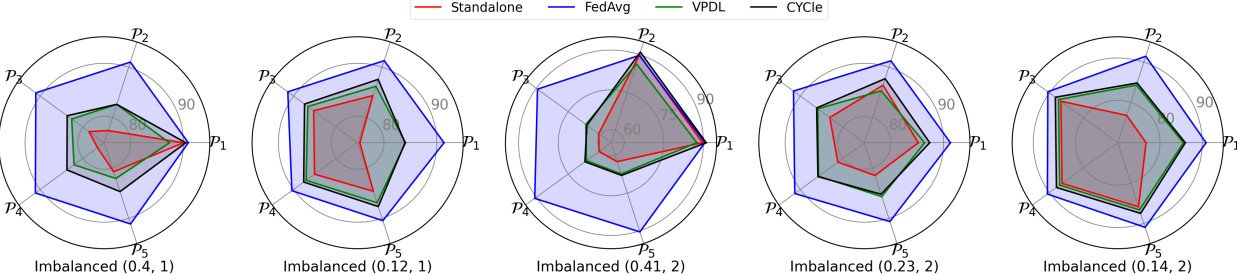

Figure 18: Per-participant performance comparison on CIFAR-10 dataset using the given imbalanced splitting scenarios.

effectiveness of our proposed CYCle method in ensuring positive collaboration gains for each participant, showcasing a higher degree of fairness in comparison to FedAvg and VPDL. For a comprehensive view of these results, please see Figure 18 and Table 10. Table 10 reports performance across five different imbalance settings using three metrics: MVA, MCG, and CGS. FedAvg suffers from a higher CGS and, in some scenarios, a large variance in individual participant outcomes, indicating fairness issues. VPDL offers stronger privacy guarantees but significantly underperforms in both accuracy and fairness. CYCle, in contrast, achieves a compelling balance. It consistently improves fairness over both baselines, as evidenced by its lower CGS across all five settings. This suggests that CYCle more equitably distributes collaborative benefits.

These findings confirm that CYCle not only preserves collaboration benefits but also effectively mitigates the challenges of unequal data distributions, an essential feature for practical decentralized learning algorithms.

### G.6 Free Rider Study

In Table 11 and Figure 19, we examine the outcomes using our CYCle approach under the varying degrees of label flipping (ranging from 0.0 to 1.0) when two participants ($\mathcal{P}_4$ and $\mathcal{P}_5$) are free riders. The findings show that the honest participants ($\mathcal{P}_1$, $\mathcal{P}_2$, and $\mathcal{P}_3$) maintain positive collaboration gains, despite the presence of malicious users / free riders. It's observed that when the flip rate is greater than 0, the collaboration gain decreases relative to scenarios where the flip rate is 0. This decline is linked to the fact that these three participants collectively hold only 60% of the total dataset. In this experiment, we use the CIFAR-10 dataset and a homogeneous data splitting approach, with other parameters remaining consistent with those detailed in Section 5.1.

Table 11: Validation accuracies of participants ($\mathcal{P}_1 - \mathcal{P}_5$) with $\mathcal{P}_4$ and $\mathcal{P}_5$ having varying rates of label flipping. The last column refers to the average accuracy of honest participants ($\mathcal{P}_1 - \mathcal{P}_3$).

| Setting | $\mathcal{P}_1$ | $\mathcal{P}_2$ | $\mathcal{P}_3$ | $\mathcal{P}_4$ | $\mathcal{P}_5$ | avg |
|---|---|---|---|---|---|---|
| Standalone ($\mathcal{B}_n$) | 83.42 | 83.67 | 83.15 | 83.13 | 83.63 | 83.41 |
| Flip rate = 0.0 | 86.12 | 87.03 | 86.33 | 85.62 | 86.10 | 86.49 |
| Flip rate = 0.2 | 84.25 | 84.20 | 84.10 | 63.10 | 65.40 | 84.18 |
| Flip rate = 0.5 | 84.30 | 84.00 | 83.95 | 38.95 | 35.75 | 84.08 |
| Flip rate = 1.0 | 84.15 | 84.20 | 83.45 | 3.40 | 3.10 | 83.93 |

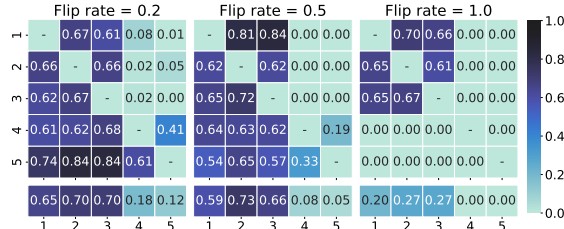

Figure 19: Heatmap visualization of reputation scores with corrupted labels at $\mathcal{P}_4$ and $\mathcal{P}_5$.

### G.7 Scalability to Larger Client Populations

To evaluate the applicability of CYCle beyond cross-silo federated learning, we conducted additional experiments on CIFAR-10 and CIFAR-100 with $N = 20$ clients. Data was partitioned using a Dirichlet distribution with $\alpha = 0.5$ to simulate non-i.i.d. heterogeneity across clients. We compare three methods: standalone (no collaboration), Gossip-SGD (with full connectivity), and CYCle.

Figure 20 displays the per-participant accuracies on CIFAR-10 and CIFAR-100 datasets. In particular, clients $2, 7$, and $15$ exhibit significantly lower standalone performance (Figure 20a) – indicating limited data quality or quantity. CYCle is able to identify such low-contributing clients through a reputation mechanism and adjusts collaboration accordingly. This ensures that while these clients benefit from collaboration, they do not disproportionately influence others, thus maintaining fairness and robustness.

These results demonstrate that CYCle maintains robust fairness and stable performance when scaled to larger client populations beyond the cross-silo setting. In contrast to Gossip-SGD, which exhibits a degradation of fairness (indicated by a negative correlation in Table 12), CYCle achieves a significantly more equitable distribution of collaborative gains.

For CIFAR-10, both methods achieve similar MCG values (28.59 for Gossip-SGD vs. 27.87 for CYCle), indicating comparable average collaborative benefits. However, the collaboration gain spread (CGS) is considerably lower for CYCle (6.06 vs. 7.70), showing that it reduces the inequality in the results of

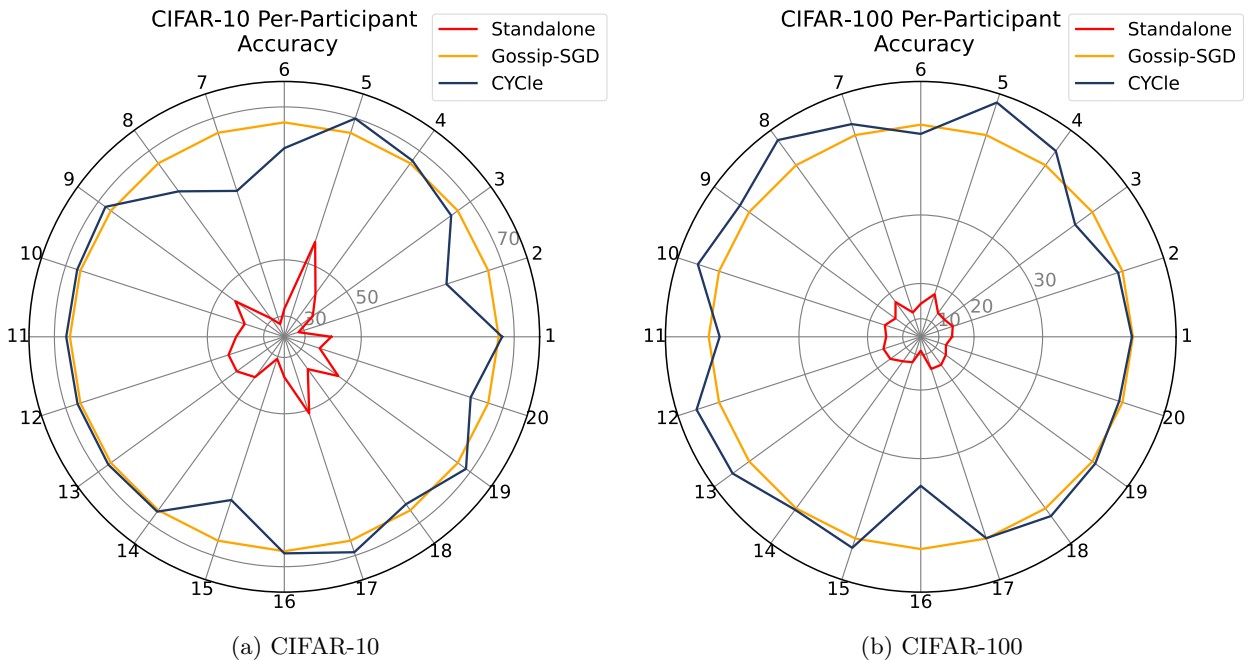

(a) CIFAR-10  (b) CIFAR-100

Figure 20: Per-participant performance radar plots for $N = 20$ clients on (a) CIFAR-10 and (b) CIFAR-100 datasets using three methods: standalone (no collaboration), Gossip-SGD (with a complete graph), and CYCle.

collaboration. Moreover, the strong positive correlation ($\rho = 0.9353$) under CYCle implies that participants with more useful or valuable data tend to receive proportionally higher rewards, signaling a fairness-aligned reward mechanism. In contrast, Gossip-SGD shows a negative correlation ($\rho = -0.4637$), which implies misalignment between client contribution and received gains.

A similar trend holds for CIFAR-100. The MCG values are again comparable, but CYCle achieves a markedly lower CGS, indicating greater consistency in how benefits are distributed. Pearson's correlation improves dramatically from $\rho = -0.7297$ (Gossip-SGD) to $\rho = 0.8164$ (CYCle), further supporting that CYCle not only promotes fairness, but aligns collaboration rewards more effectively with actual client contribution.

We also note that CYCle can be naturally extended to scenarios with significantly larger client populations and partial participation. In such cases, a subset of clients can be sampled in each round and CYCle can be applied by constructing a subgraph over the active clients. This involves computing a sub-mixing matrix using only the participating clients and running the CYCle protocol locally on this induced set of clients. Reputation and gradient alignment remain local and round-specific, requiring no global state. This design makes CYCle compatible with the standard partial participation strategies commonly used in cross-device FL. However, since our focus is on cross-silo settings with full participation, we leave empirical evaluation under partial participation for future work.

| Dataset | Method | MCG | CGS | Pearson corr. ($\rho$) |
|---|---|---|---|---|
| CIFAR-10 | Gossip-SGD | 28.59 | 7.70 | -0.4637 |
| | CYCle | 27.87 | **6.06** | **0.9353** |
| CIFAR-100 | Gossip-SGD | 22.68 | 2.12 | -0.7297 |
| | CYCle | 22.88 | **1.25** | **0.8164** |

Table 12: Comparison of collaborative fairness metrics for $N = 20$ clients.

In summary, CYCle preserves average performance while significantly improving fairness and reward alignment between participants. These gains are particularly important in federated settings with data heterogeneity and varying client quality, as they help prevent both under- and over-compensation in collaborative learning.

## G.8 Trade-off between MCG and CGS

Figure 21 presents a plot depicting the MCG and CGS values for 10 different data-split scenarios, which are specified in Table 1 and Table 10. Each point in the plot corresponds to the average performance of a method under one scenario, with the $x$-axis representing the mean collaboration gain (MCG) and the $y$-axis representing the collaboration gain spread (CGS). The bottom right region of the plot, indicating high MCG and low CGS, represents the ideal operational regime, where all participants benefit meaningfully and fairly from collaboration. It must be emphasized that the results of our CYCle approach fall predominantly towards the bottom-right corner of the plot. On average, CYCle achieves the highest MCG and the lowest CGS in the 10 scenarios. While baseline methods such as CFFL, RFFL, and CGSV may outperform CYCle in one of the two metrics under certain scenarios, they exhibit a clear trade-off: improvements in collaborative gain often come at the cost of high inequality (larger CGS) and vice versa. For example, CGSV may occasionally achieve a higher MCG, but its CGS is substantially worse, indicating that a few participants benefit disproportionately, while others gain little or even suffer negative outcomes. In contrast, methods such as RFFL may better control CGS but fail to deliver sufficient gain, as reflected in lower MCG values.

The directionality of the arrows overlaid on the figure further emphasizes the optimization goal: to move toward the bottom right.

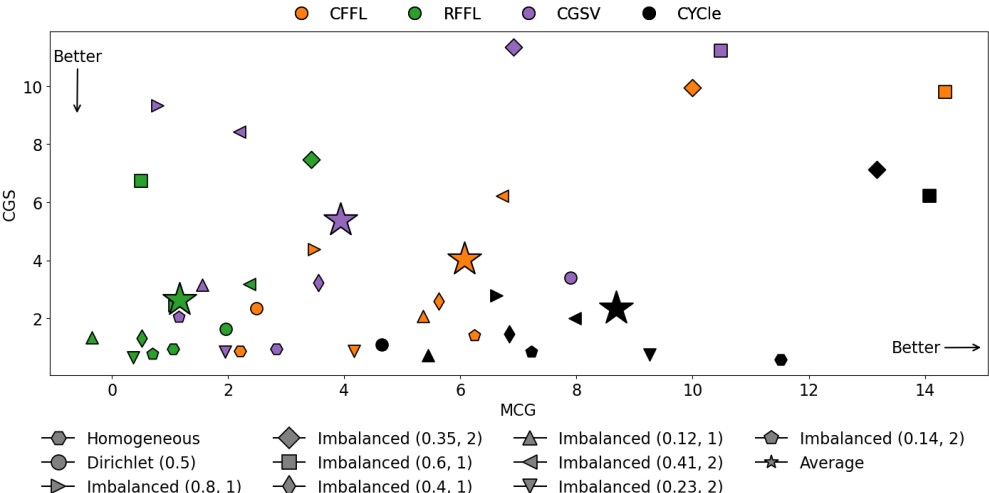

Figure 21: Plot of MCG and CGS values for 10 distinct data splits, ranging from homogeneous to a spectrum of imbalanced scenarios.

## G.9 Hyperparameter Sensitivity

**CYCle (PDL).** We have defined only two hyperparameters in our model, $\tau_{opt}$ and $\tau_{max}$, which have a geometric interpretation (degree of gradient alignment required for beneficial collaboration). In our experiments, we set $\tau_{opt} = 0.25$ and $\tau_{max} = 0.75$. Figure 22a shows the sensitivity analysis for these hyperparameters, where it is evident that increasing the values of $\tau_{max}$ and $\tau_{opt}$ correlates with improved MCG in a homogeneous setting. This is expected as higher values of these parameters bring our algorithm closer to the VPDL framework. However, the MCG and CGS (Figure 22b) values for different hyperparameter settings are comparable, indicating that the reported results are robust.

These parameters play a well-defined and interpretable role in our framework through the mapping function (Equation 7). The parameters $\tau_{opt}$ and $\tau_{max}$ define a soft acceptance region over the similarity space (e.g.,

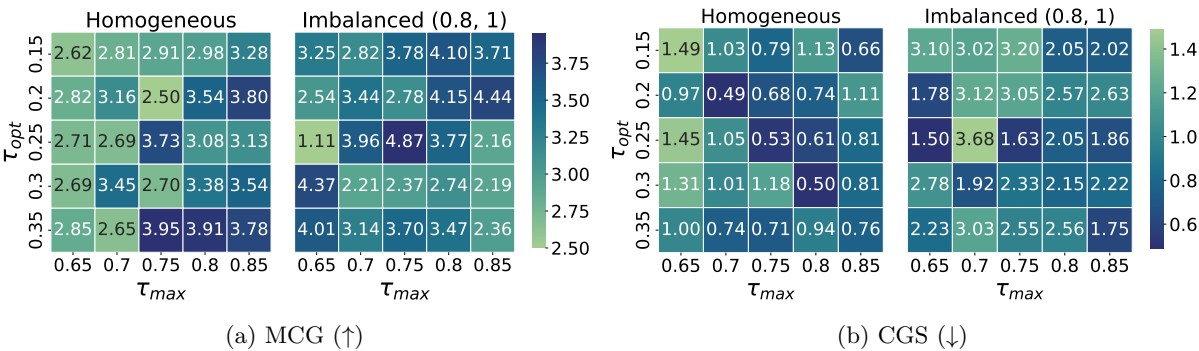

(a) MCG (↑)

(b) CGS (↓)

Figure 22: Sensitivity analysis to $\tau_{opt}$ and $\tau_{max}$ parameters.

give full credit to highly aligned clients $h(s) = 1$, while blocking collaboration with poorly aligned ones $h(s) = 0$). As shown in Figure 22, we explored a range of reasonable values and found that going below $\tau_{opt} = 0.25$ results in degraded collaboration fairness and utility, as unqualified updates are included too permissively. The selected values are therefore informed by both the empirical similarity statistics and the theoretical geometry of the collaboration space. For these reasons, we did not conduct exhaustive tuning and instead used principled, task-informed choices.

**CYCle (Gossip-SGD).** The only hyperparameter in the gossip-based variant of our protocol is $\beta$, which controls the sharpness of the softmax function used to derive the communication probabilities from gradient alignment scores. Higher $\beta$ values assign greater weight to stronger alignments, enforcing more selective and fairness-oriented peer interactions, while lower values yield more uniform communication, resembling a fully connected topology. Figure 23a shows the performance across participants under varying $\beta \in \{5, 15, 35, 50, 100\}$. Although participant-level performance varies slightly, all configurations exhibit reasonable fairness, validating the robustness of CYCle under different settings. Figure 23b further illustrates the trade-off between utility (MCG) and fairness (Pearson correlation of pre- and post-collaboration accuracies). We observe that intermediate values of $\beta$ (e.g., 35-50) strike a favorable balance, delivering high collaboration gain without sacrificing fairness.

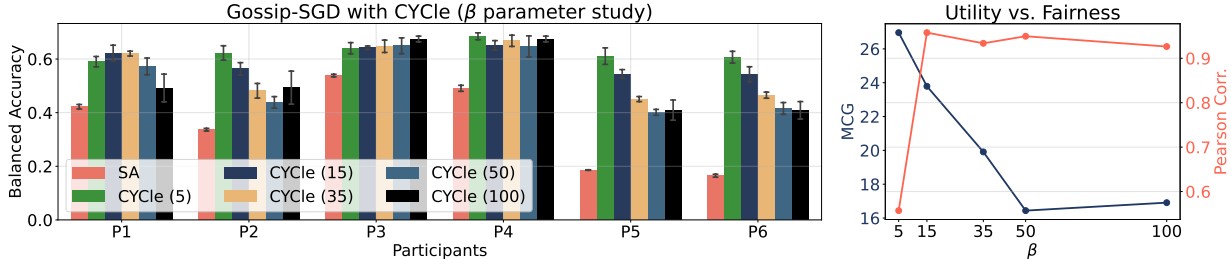

(a) Performance of CYCle under different $\beta$ parameters $\in \{5, 15, 35, 50, 100\}$. (b) MCG vs. correlation coefficient.

Figure 23: Study on the impact of parameter $\beta$ on performance and analysis of the utility versus fairness trade-off using the CYCle algorithm.

## G.10 Additional Results

The accuracy metrics for the individual participants for the experiments on the CIFAR-10 dataset (corresponding to Figures 3 and 4) are summarized in Table 13 for greater clarity. These results clearly demonstrate the negative gain experienced by the participant $\mathcal{P}_1$ in the imbalanced settings under most existing FL algorithms and how the proposed CYCle approach overcomes this problem.

Table 13: Per-participant performance comparison on CIFAR-10 dataset: Validation accuracy evaluated with $N = 5$ participants. The left table presents the performance of our proposed framework compared to the FedAvg algorithm. The right table compares collaboration gain and fairness of our proposed CYCle algorithm with existing works (columns 4-6). We employ MVA ($\uparrow$), MCG ($\uparrow$) and CGS ($\downarrow$) as evaluation metrics.

| Split | No./Metric | SA | FedAvg | VPDL | CYCle |
|---|---|---|---|---|---|
| Homogeneous | 1 | 83.42 | 91.02 | 85.82 | 86.12 |
| | 2 | 83.67 | 90.32 | 84.02 | 87.03 |
| | 3 | 83.15 | 90.63 | 84.70 | 86.33 |
| | 4 | 83.13 | 90.82 | 85.27 | 85.62 |
| | 5 | 83.63 | 90.23 | 85.12 | 86.10 |
| | MVA | 83.40 | 90.60 | 84.98 | 86.24 |
| | MCG | 0.00 | **7.20** | 1.58 | 2.84 |
| | CGS | 0.00 | 0.48 | 0.71 | **0.37** |
| Dirichlet (0.5) | 1 | 68.47 | 88.92 | 74.47 | 79.48 |
| | 2 | 65.25 | 88.48 | 74.17 | 73.55 |
| | 3 | 73.92 | 88.35 | 77.17 | 79.87 |
| | 4 | 67.55 | 88.68 | 74.17 | 78.70 |
| | 5 | 61.62 | 89.35 | 71.40 | 73.05 |
| | MVA | 67.36 | 88.76 | 74.27 | 76.93 |
| | MCG | 0.00 | **21.40** | 6.91 | 9.57 |
| | CGS | 0.00 | 4.31 | 2.31 | **2.13** |
| Imbalanced (0.8, 1) | 1 | 92.77 | 90.23 | 89.87 | 93.80 |
| | 2 | 56.85 | 90.33 | 60.77 | 63.02 |
| | 3 | 53.82 | 90.07 | 62.08 | 62.82 |
| | 4 | 58.00 | 90.15 | 61.52 | 63.35 |
| | 5 | 57.68 | 90.07 | 61.68 | 63.30 |
| | MVA | 63.82 | 90.17 | 67.18 | 69.26 |
| | MCG | 0.00 | **26.35** | 3.36 | 5.44 |
| | CGS | 0.00 | 14.51 | 3.58 | **2.56** |
| Imbalanced (0.35, 2) | 1 | 89.30 | 90.45 | 85.50 | 89.90 |
| | 2 | 88.98 | 90.53 | 86.00 | 89.73 |
| | 3 | 69.07 | 89.83 | 73.97 | 75.35 |
| | 4 | 69.23 | 90.40 | 74.53 | 75.58 |
| | 5 | 69.53 | 90.48 | 73.55 | 75.02 |
| | MVA | 77.22 | 90.34 | 78.71 | 81.12 |
| | MCG | 0.00 | **13.12** | 1.49 | 3.89 |
| | CGS | 0.00 | 9.61 | 4.01 | **2.65** |
| Imbalanced (0.6, 1) | 1 | 92.27 | 90.38 | 89.25 | 92.80 |
| | 2 | 68.65 | 90.73 | 71.58 | 71.80 |
| | 3 | 69.12 | 89.92 | 71.53 | 72.65 |
| | 4 | 68.78 | 89.33 | 72.85 | 73.45 |
| | 5 | 70.60 | 90.43 | 71.95 | 72.87 |
| | MVA | 73.88 | 90.16 | 75.43 | 76.72 |
| | MCG | 0.00 | **16.28** | 1.55 | 2.83 |
| | CGS | 0.00 | 9.11 | 2.45 | **1.38** |

| Split | No./Metric | SA | CFFL | RFFL | CGSV | CYCle |
|---|---|---|---|---|---|---|
| Homogeneous | 1 | 60.98 | 62.05 | 61.65 | 63.32 | 72.42 |
| | 2 | 59.87 | 63.60 | 62.32 | 63.67 | 70.88 |
| | 3 | 59.20 | 61.58 | 61.07 | 63.08 | 71.48 |
| | 4 | 60.50 | 62.50 | 60.80 | 61.88 | 72.55 |
| | 5 | 61.67 | 63.53 | 61.65 | 64.42 | 72.43 |
| | MVA | 60.44 | 62.65 | 61.50 | 63.27 | 71.95 |
| | MCG | 0.00 | 2.21 | 1.05 | 2.83 | **11.51** |
| | CGS | 0.00 | 0.87 | 0.95 | 0.94 | **0.58** |
| Dirichlet (0.5) | 1 | 48.17 | 47.65 | 48.45 | 55.30 | 51.78 |
| | 2 | 48.72 | 49.28 | 50.07 | 53.10 | 53.20 |
| | 3 | 49.15 | 51.15 | 50.70 | 53.60 | 53.82 |
| | 4 | 44.17 | 49.30 | 45.72 | 54.70 | 47.95 |
| | 5 | 42.57 | 47.82 | 47.65 | 55.57 | 49.28 |
| | MVA | 46.55 | 49.04 | 48.52 | 54.45 | 51.21 |
| | MCG | 0.00 | 2.49 | 1.96 | **7.90** | 4.65 |
| | CGS | 0.00 | 2.35 | 1.63 | 3.40 | **1.11** |
| Imbalanced (0.8, 1) | 1 | 74.32 | 69.38 | 71.47 | 56.55 | 77.15 |
| | 2 | 52.95 | 56.08 | 52.80 | 56.92 | 57.10 |
| | 3 | 48.95 | 55.52 | 52.13 | 54.77 | 58.73 |
| | 4 | 48.40 | 54.97 | 52.40 | 55.33 | 57.93 |
| | 5 | 51.30 | 57.35 | 52.53 | 56.23 | 58.08 |
| | MVA | 55.18 | 58.66 | 56.27 | 55.96 | 61.80 |
| | MCG | 0.00 | 3.48 | 1.08 | 0.78 | **6.62** |
| | CGS | 0.00 | 4.39 | **2.45** | 9.32 | 2.79 |
| Imbalanced (0.35, 2) | 1 | 69.43 | 66.38 | 64.22 | 63.10 | 74.13 |
| | 2 | 69.70 | 68.52 | 63.55 | 62.15 | 74.07 |
| | 3 | 45.17 | 62.18 | 53.70 | 60.58 | 64.93 |
| | 4 | 44.02 | 63.93 | 54.12 | 61.53 | 63.93 |
| | 5 | 47.12 | 64.37 | 56.98 | 62.60 | 64.23 |
| | MVA | 55.09 | 65.08 | 58.51 | 61.99 | 68.26 |
| | MCG | 0.00 | 9.99 | 3.43 | 6.91 | **13.17** |
| | CGS | 0.00 | 9.95 | 7.46 | 11.34 | **7.12** |
| Imbalanced (0.6, 1) | 1 | 74.23 | 69.30 | 61.58 | 62.33 | 76.22 |
| | 2 | 45.08 | 65.70 | 49.05 | 61.43 | 64.17 |
| | 3 | 46.08 | 62.05 | 47.15 | 60.40 | 61.88 |
| | 4 | 43.80 | 63.97 | 48.67 | 60.53 | 59.10 |
| | 5 | 45.05 | 64.98 | 50.30 | 61.98 | 63.30 |
| | MVA | 50.85 | 65.20 | 51.35 | 61.34 | 64.93 |
| | MCG | 0.00 | **14.35** | 0.50 | 10.49 | **14.08** |
| | CGS | 0.00 | 9.79 | 6.74 | 11.23 | **6.22** |

