# OpenReview forum: "CYCle: Choosing Your Collaborators Wisely to Enhance Collaborative Fairness in Decentralized Learning"
_TMLR — Accepted by TMLR_

### Review · Reviewer_XGxa · 2025-05-10

**Summary Of Contributions:**

This article presents the **CYCle protocol**, a reputation scoring mechanism based on gradient alignment, aimed at improving collaborative fairness in private decentralized learning (PDL). The core contributions include:
1. Highlighting the shortcomings of traditional collaborative fairness metrics (based on the correlation between contribution and final accuracy) in scenarios of negative collaboration gain, a new paradigm for quantifying fairness is proposed, focusing on maximizing mean collaboration gain (MCG) and minimizing collaboration gain spread (CGS).
2. A dynamic reputation scoring mechanism is designed, which assesses participant contribution through gradient alignment of cross-entropy loss and distillation loss, and based on this, an adaptive knowledge sharing strategy is implemented to ensure that all participants achieve positive collaboration gains.
3. The CYCle protocol is extended to decentralized algorithms such as Gossip-SGD, achieving fair collaboration through a dynamic mixing matrix, and proving its superiority to FedAvg in heterogeneous data scenarios.
4. The effectiveness of the method is validated on datasets such as CIFAR-10, CIFAR-100, and Fed-ISIC2019, especially demonstrating a significant reduction in CGS in highly non-independent and identically distributed (Non-IID) data scenarios, while maintaining high MCG.

**Audience:**

Yes

**Broader Impact Concerns:**

1. **Privacy Protection Measures**: Although CaPriDe uses fully homomorphic encryption, it is necessary to clarify the specific protection mechanisms for sensitive information in the training data (e.g., medical images from Fed-ISIC2019).
2. **Abuse Risk Analysis**: Discuss the potential risks of the model being misused for unfair competition (e.g., high-contribution participants monopolizing knowledge sharing) and mitigation strategies.

**Claims And Evidence:**

Yes

**Requested Changes:**

**Key Modifications**
1. **Supplement Comparative Experiments**: Include comparisons with recent SOTA methods (such as FedFair [1], FairVFL [2]) to validate the superiority of CYCle in a broader range of scenarios, which impacting the validity of conclusions.
2. **Add Complexity Analysis**: Supplement time complexity and communication overhead data in Tables or the appendix (such as single-round training time, encryption operation time).

**Improvement Suggestions**
1. **Expand Adversarial Experiments**: Increase robustness testing against attacks such as gradient tampering and model poisoning (Section 5.2).
2. **Parameter Sensitivity Analysis**: Provide a quantitative analysis of the impact of τ_opt and τ_max on MCG/CGS (currently only partially shown in Figure 20).

[1] Che, X., Hu, J., Zhou, Z., Zhang, Y., and Chu, L., “Training Fair Models in Federated Learning without Data Privacy Infringement”, <i>arXiv e-prints</i>, Art. no. arXiv:2109.05662, 2021. doi:10.48550/arXiv.2109.05662.

[2] Qi, T., “FairVFL: A Fair Vertical Federated Learning Framework with Contrastive Adversarial Learning”, <i>arXiv e-prints</i>, Art. no. arXiv:2206.03200, 2022. doi:10.48550/arXiv.2206.03200.

**Strengths And Weaknesses:**

**Strengths**
1. **Methodological Innovation**: Introduced gradient alignment into reputation scoring for the first time, combining distillation loss to dynamically adjust knowledge sharing weights, providing a new perspective on decentralized collaborative fairness (see Section 4.1).
2. **Theoretical Rigor**: Through theoretical analysis of the mean estimation problem, demonstrated the superiority of CYCle under data heterogeneity (Section 4.3 and Appendix A), enhancing the credibility of the method.
3. **Experimental Comprehensiveness**: Covered multiple data partitioning scenarios including homogeneous, heterogeneous (Dirichlet distribution), and imbalanced data, and validated robustness across datasets (Fed-ISIC2019) and architectures (ResNet18 vs. custom CNN) (Section 5.2).
4. **Reproducibility**: Complete hyperparameter configurations (Section 5.1) provide an important benchmark for subsequent research.

**Weaknesses**
1. **Insufficient Baseline Comparison**: Did not compare with the latest SOTA methods (such as FedFair [1], FairVFL [2]), which weakens the universality of the conclusions (Tables 1 and 2).
2. **Lack of Complexity Analysis**: Did not quantify computational costs (such as the time for computing encrypted distillation loss) and communication costs (such as the dynamic mixing matrix update frequency in Gossip-SGD extensions), affecting the practical assessment (Section 5.1).
3. **Insufficient Adversarial Robustness**: Only simulated label-flipping attacks (Section 5.2) and did not consider more complex Byzantine attacks (such as gradient tampering), limiting the method's application potential in open environments.

[1] Che, X., Hu, J., Zhou, Z., Zhang, Y., and Chu, L., “Training Fair Models in Federated Learning without Data Privacy Infringement”, <i>arXiv e-prints</i>, Art. no. arXiv:2109.05662, 2021. doi:10.48550/arXiv.2109.05662.

[2] Qi, T., “FairVFL: A Fair Vertical Federated Learning Framework with Contrastive Adversarial Learning”, <i>arXiv e-prints</i>, Art. no. arXiv:2206.03200, 2022. doi:10.48550/arXiv.2206.03200.

---

> ### Author Response · Authors · 2025-06-04
>
> We sincerely thank Reviewer XGxa for their thoughtful and encouraging feedback. We greatly appreciate your recognition of the methodological innovation introduced by integrating gradient alignment with reputation scoring, as well as your acknowledgment of the theoretical rigor, comprehensive experimental evaluation, and attention to reproducibility. Your detailed comments affirm the core contributions of our work and motivate us to further refine and strengthen our approach. Below, we address your suggestions and clarify key points raised in your review.
>
> **W1. Insufficient Baseline Comparison.** Thank you for highlighting the baseline-comparison issue and giving us the chance to clarify our choices.
> - FedFair [1] targets *predictive/group fairness* of the global model. Our work targets *collaborative fairness* - fair benefit allocation in federated training. The objectives and metrics are therefore incomparable.
> - FairVFL [2] is built for *vertical FL*, whereas our study addresses *horizontal FL*; the two settings use incompatible system assumptions.
> - We therefore, to the best of our knowledge, benchmarked against the state-of-the-art collaborative-fairness methods in horizontal FL (CFFL, RFFL, CGSV) and standard FL baselines, which are relevant comparators for our problem.
>
> **W2. Lack of Complexity Analysis.** Thank you for pointing out the need to clarify the computational and communication complexities. We have now addressed this concern in the revised version of the manuscript. Please refer to **Appendix E** (titled Computational and Communication Complexity), where we provide the requested details. All relevant additions and clarifications are **highlighted in blue** for ease of reference.
>
> **W3. Insufficient Adversarial Robustness.** Thank you for the thoughtful suggestion regarding adversarial robustness. Our work primarily focuses on **collaborative fairness** in federated learning. The label-flipping scenario in Section 5.2 was included to illustrate the behavior of a free-rider client, one with systematically poor data quality, rather than to model a full-fledged Byzantine adversary. While robustness to Byzantine attacks is a distinct and important topic beyond the scope of this work, we note that our method is designed to penalize unhelpful or malicious updates, which can naturally downweight adversarial clients. A deeper investigation of such robustness is a valuable direction for future work.
>
> **IS2. Parameter Sensitivity Analysis.** Thank you for your suggestion regarding the sensitivity analysis of $\tau_{opt}$ and $\tau_{max}$. We have clarified our rationale and included a more detailed discussion in the revised manuscript. Please refer to **Appendix G.9** for the quantitative analysis and interpretation of these parameters.
>
> **Broader Impact Concerns.** Thank you for raising these important concerns. We have now addressed both points in the Broader Impact Statement of the revised manuscript. Please refer to that section for clarification regarding (1) privacy protection measures using FHE, and (2) abuse risk and mitigation strategies designed to prevent knowledge monopolization.

---

> > ### Comment · Reviewer_XGxa · 2025-06-26
> > **Rebuttal Acknowledgment**
> >
> > Thank the authors for their revisions to the manuscript. All my concerns and doubts have been well addressed, but please double-check the whole paper thoroughly for any typos or irregularities before acceptance.

---

### Review · Reviewer_xUX9 · 2025-05-21

**Summary Of Contributions:**

This paper presents CYCle, a novel protocol designed to enhance collaborative fairness in privacy-preserving decentralized learning. The method introduces a gradient alignment-based reputation mechanism to dynamically adjust knowledge transfer between participants and promotes fairer outcomes across heterogeneous clients.

**Audience:**

Yes

**Claims And Evidence:**

Yes

**Requested Changes:**

See the weaknesses above.

**Strengths And Weaknesses:**

Strengths:

The paper addresses the important yet underexplored problem of fairness in decentralized collaborative learning, particularly without centralized coordination—relevant to privacy-sensitive domains such as healthcare and finance.The authors identify limitations of correlation-based fairness metrics and propose a principled alternative based on maximizing Mean Collaboration Gain while minimizing its standard deviation. This is well-justified both theoretically and empirically. The method is comprehensively evaluated on CIFAR-10, CIFAR-100, and Fed-ISIC2019 under various data heterogeneity and imbalance scenarios, demonstrating consistently positive collaboration gains and superior fairness compared to baselines.

Weaknesses:

The theoretical analysis is confined to a simple two-party mean estimation example. A more general or deeper theoretical understanding for complex models (e.g., neural networks) is lacking.

The protocol introduces non-negligible computation (e.g., gradient alignment and periodic sharing decisions). The impact on overall convergence speed and computational/communication overhead is not sufficiently analyzed.

The choice of baselines is somewhat inappropriate. While the comparison to FL algorithms is appreciated, fairness comparisons with more recent or competitive decentralized learning frameworks (e.g., Cronus) are missing.

There are minor issues in the writing. For example, the definition of $\nu$ and $\mu$ in the bound of $P(G_n < 0)$ is not clearly stated.

---

> ### Author Response · Authors · 2025-06-04
>
> We sincerely thank Reviewer xUX9 for the constructive and encouraging feedback. We are grateful for your recognition of the importance of fairness in decentralized collaborative learning, especially in privacy-sensitive settings like healthcare and finance. Your appreciation of our principled approach, grounded in optimizing mean collaboration gain and its variability, as well as your acknowledgement of our theoretical justification and comprehensive empirical validation across diverse datasets and data conditions, is deeply valued. We address your comments and suggestions in detail below.
>
> **W1. The theoretical analysis.** Thank you for raising this point. Our theoretical analysis is intentionally grounded in a simple mean estimation setting to provide a tractable and interpretable example that captures the essence of collaboration gain and its dependency on data similarity. This simplified setup allows us to formally characterize the failure of naive federated averaging under heterogeneity and derive analytical insights into when and why similarity-aware collaboration becomes necessary, insights that directly inform the design of CYCle.
>
> While we agree that extending the theory to complex models like neural networks would be valuable, such analyses are typically intractable without strong simplifying assumptions, and to the best of our knowledge, no existing work provides a rigorous theoretical treatment in neural network-based federated learning. Our work follows the common approach in ML research of pairing formal insights from simplified settings with empirical validation on realistic deep learning tasks. We clarified this motivation and limitation in the revised version.
>
> **W2. Computational & Communication Overhead.** Thank you for this valuable observation. We clarify the following:
>
> CYCle introduces minimal/no overhead in both computation and communication relative to baseline methods:
> - Gradient alignment computations involve inner products between local and received gradients, which are computed once per peer every $R$ rounds and scale linearly with model size. In practice, these operations add negligible time (${<}5$%) to each local step.
> - Sharing decisions are made periodically (every $R$ rounds) and are based on locally computed similarity scores. These decisions do not involve additional communication, as the dynamic mixing matrix is never exchanged; clients only share model updates, identical to (if everyone shares all the time) or better than standard Gossip-SGD.
> - Convergence speed is not negatively impacted, as shown in Figure 15, CYCle consistently converges within the same number of rounds as the baselines while achieving superior fairness.
>
> Further details and quantitative overhead estimates are provided in **Appendix E**.
>
> **W3. The choice of baselines.** Thank you for the suggestion. We appreciate the reviewer's interest in stronger decentralized baselines.
>
> Our primary focus is on **collaborative fairness in federated learning**, and, to the best of our knowledge, we included the most relevant fairness-aware FL methods in this setting, namely CFFL, RFFL, and CGSV, alongside standard FL and PDL baselines. These are widely recognized as state-of-the-art for fairness evaluation. To the best of our knowledge, CYCle is the first to explicitly address collaborative fairness in decentralized learning, whereas existing decentralized learning frameworks have not considered this aspect.
>
> While Cronus is a robust and decentralized collaborative framework, it is primarily designed to defend against privacy and poisoning attacks by reducing the dimensionality of exchanged information and treating models as black boxes. Its primary goal is robustness and security, not fairness in collaboration or equitable gain allocation. Cronus operates under different assumptions (e.g., adversarial environments and heterogeneous model architectures) and does not account for contribution estimation or fairness tracking across participants.
>
> We now clarify this positioning explicitly in the revised manuscript:
> - In Section 2, we note that Cronus focuses on robustness and does not target fairness,
> - In Section 5.1, we explain our rationale for selecting fairness-aware baselines over robustness-centric ones.
>
> We agree this is a promising direction and note in the paper that integrating fairness into robustness-oriented decentralized frameworks like Cronus is left for future work.
>
> **W4. Minor issue.** Thank you for pointing this out. We would like to clarify that the definitions of $\nu$ and $\mu$ are indeed provided in the last paragraph on page 5. To improve clarity, we revised the paragraph slightly in the final version to make these definitions more explicit.

---

> > ### Comment · Reviewer_xUX9 · 2025-06-27
> >
> > Thanks to the authors for their reply, I have no further questions.

---

### Review · Reviewer_5Hrc · 2025-05-30

**Summary Of Contributions:**

This submission studies the problem of collaborative fairness in decentralized learning, i.e. designing a learning system that fairly rewards each involved party from the perspective of their perfomance gains derived by participating to the collaboration. This is opposed to just maximizing the expected accuracy gain, which can be not fair for each individual party. To this end, they propose novel fairness objective that maximize the "mean collaboration gain" (MCG) while minimizing the "collaborative gain spread" (CGS), and propose the CYCle protocol for private decentralized learning (PDL). Authors also show that their protocol can be adapted to work with gossip-based decentralized algorithms. The paper includes theoretical results lower bounding the probability of achieving better results than FedAvg on a simple problem with 2 clients, and presents wide experimentation on CIFAR-10/100 and Fed-ISIC, in both homogeneous and heterogeneous settings. Overall, the paper proves quite a strong point in motivating the need of a different collaborative metric, and results clearly support the claims.

**Audience:**

Yes

**Claims And Evidence:**

Yes

**Requested Changes:**

## Critical changes
1. **Revise the placement of contents:** in my opinion, the theoretical part and the extension to Gossip-SGD are important contribution and should appear in the main paper. In particular, section A (theoretical contribution) should be moved back in the main paper in section 4.3 and section C to section 4.4. At the same time, the implementation details of section 5.1 (which occupy about a page) can be moved to the appendix. Some sections (e.g. sec. 3) could me more concise. If space is an hard constraint (e.g. if the paper has to fit 12 pages of main content) I believe figure 4 could be moved to the appendix and figures 2-7 reduced in sized or moved as well. Results on FedIsic (section D.3) are explicitly claimed and should be presented in the main paper.
2. **Discusss more in detail the limitation of theoretical results:**  If my intuition is correct, the guarantees of theorem A.1 only hold for the specific problem considered, and does not trivially extend to other cases. Can authors confirm if this is the correct interpretation? If yes, I think it would be beneficial to discuss the current limitation, such that future work may improve. For example, would the findings generalize with arbitrarily as many clients? If yes, do you believe it is possible to provide a more general proof? If not, could you indicate which are the difficulties in proving a more general claim?

## Suggested changes
1. **Include details about communication cost:** while it is not the focus of the paper, I think reporting how much data has been exchanged during the training process can be useful for future research on how to make the learning process more communication efficient.

## Minor
1. **Typos:** There could be some typos in the proof of Lemma A.1: I believe that second line of eq. (17) misses the application of similar passages applied to the first line, and reference to eq. (14) should probably refer to eq. (15) instead.
2. **Notations:** the notation (sec 2.1) is consistent and mostly clear, but I found some difficulties recalling the meaning of each symbol in later sections of the paper. I invite the authors to consider simplifying it for improving clarity.

## Other suggestions / Questions
1. I would briefly restate the advantages of the proposed approach w.r.t. previous ones just before section 2.1
2. **Suggestion on experimental setting:** I would have considered datasets with severe domain shift for the experimentation (e.g. DomainNet): in those cases, if my understanding is correct, the algorithm would streghten the reputation scores of clients holding examples from the same domain. Probably, the algorithms would have worked even better in those cases (see next point for rationale).
3. **Connection to PFL:** If we interpret the matrix of reputations scores as the adiancency matrix of a graph, the algorithm could be used to find cliques of clients with similar objectives, and allow more general forms of personalization than letting each client have a (potentially) different model than others. Did you think of possible extensions in this sense?
4. **Intuitions about extension to other tasks?** Can you share some intuitions of what you think we can expect applying the proposed scheme in NLP?

**Strengths And Weaknesses:**

## Strenghts
1. **Well motivated research problem and rationale:** The problem tackled in this paper is important, as authors very clearly show by examples in section 3. The rationale behind their proposed approach is conceptually simple and appears as a clean problem reformulation w.r.t the classical global risk minimization common in general decentralized optimization.
2. **Nice theoretical foundations:** authors include a lower bound that proves an advatage w.r.t FedAvg in learning in a fair manner. Even though the settings is somewhat synthetic and of limited general utility (see questions on that), is provides solid grounding to the proposed methodology.
3. **Good empirical results:** besides the "goodness" of the results per se, the experiments directly validate the claims on the proposed approach, specifically the fact that the algorithm automatically "discovers" which clients are useful for their own objective and which are not.


## Weaknesses
1. **Placement of the contents:** in my opinion, this paper contains more than 12 pages of "actual" main content. I believe some choices regarding the placement of contents between main paper and appendix do not fully expose the full contribution, as I required reviewing significant portion of the appendix, while some sections of the main paper could go to the appendix. Addressing this concern is very important to ensure the clarity of the final manuscript. Please have a look to the requested changes for some suggestions.
2. **Unclear motivation behind metric clipping and momentum in score calculation:** misalignment metric clipping is motivated by experimental evidence, but it is not evident which is the role of the involved hyperparameters on the fairness objective. Similarly, momentum in score calculation is used as a way to smooth it down: it is not clear to me why we would expect large stochastic variations of the reputation score, especially in later stages of training. I would like the authors to discuss these aspects, and if possible suggest more straighforward altenatives.
3. **It is unclear if the approach is limited to cross-silo FL settings:** the experiment only consider cross-silo settings with very low number of clients. A motivation for that is the burden required for PDL, which restricts the feasibility to cases in which the number of clients is small. However, it is not clear to me is the approach per se would work equally well on settings with more clients, and possibly partial participation. I understand this is somewhat outside the scope, but it would be nice if authors could discuss this aspect and possibly provide some experiments with larger number of clients.
4. **Unclear extent of theoretical claims:** it is unclear to me wether the theoretical part, which is appreciated, has some predictive value for real-world cases outside the toy theoretical setting with 2 clients. This doubt arise from the fact that, while lower bounds are usually on minimum errors (_i.e._ proving that there exist at least one problem/family of problems in which a given measure of error is $\geq \epsilon > 0$), in this work is used to provide the minimum probability of __success__ only for the learning problems of the form of theorem A.1. In that sense, while lower bounds are typically strong, in this case the result is limited to a very specific setting. The claim in the paper is correct (_i.e._ it is reported that the guarantees hold for that specific setting), but I think that the lack of discussion could mislead readers into thinking that the result have more general validity. I believe that moving section A into the main paper (see requested critical changes), would give opportunity to remark the extent of the theoretical claims.

---

> ### Author Response · Authors · 2025-06-04
> **Official Comment by Authors (1/2)**
>
> We sincerely thank Reviewer 5Hrc for the thoughtful and encouraging review, as well as for the insightful suggestions for future work. We greatly appreciate your recognition of the strong motivation behind our research problem and the clarity of our conceptual reformulation relative to traditional global risk minimization. We are also grateful for your acknowledgment of the theoretical foundations we provide, particularly the fairness-aware lower bound comparison with FedAvg, as well as your positive assessment of our empirical results and their alignment with our key claims. Your feedback helps validate our contributions and guides further refinement of our work, as we address your comments in detail below.
>
> **W1. Placement of the contents.** Thank you for this detailed and constructive recommendation. We fully agree with the reviewer's assessment regarding the placement of key content and have revised the manuscript accordingly to improve clarity and emphasis on our core contributions.
>
> **W2. Motivation behind metric clipping and momentum.** Thank you for raising these important questions regarding the motivation for misalignment metric clipping and momentum in the reputation score computation.
>
> The clipping of similarity scores (via $\tau_{opt}$ and $\tau_{max}$) serves as a soft thresholding mechanism to bound the influence of peers with highly misaligned objectives. This is not just empirically motivated but also aligns with our fairness objective: unbounded similarity scores can overly reward or penalize certain clients, which is critical for fair benefit estimation.
>
> Momentum is introduced in the update of the reputation score to reduce noise from round-to-round fluctuations, especially when client updates are affected by mini-batch stochasticity and local non-convexity during training. While these factors directly impact parameter updates or gradients, they indirectly influence the directional similarity used in score computations. As such, applying momentum helps smooth the reputation score over time, especially in the early stages of training or when the update frequency $R$ is small (e.g., $R=1$), where decisions are based on limited information from fewer gradient steps, and thus are more prone to high variance estimates. On the other hand, when $R$ is sufficiently large (e.g., $R=5$), the system implicitly averages over more local steps, at which point the need for the momentum term diminishes.
>
> An alternative to using momentum is to simply use the raw similarity scores, without any smoothing. This approach becomes more viable when using a larger update interval $R$.
>
> **W3. It is unclear if the approach is limited to cross-silo FL settings.** Thank you for raising this point. The CYCle protocol itself is not inherently limited to small-scale or full participation setups. Since the protocol relies only on local similarity computation and peer-to-peer update exchanges, it can naturally be extended to larger populations using full/partial participation.
>
> To demonstrate this, we have conducted additional experiments with $N=20$ clients on CIFAR-10 and CIFAR-100, using Dirichlet partitioning with $\alpha=0.5$ to simulate realistic non-IID settings. The results, shown in the radar plots and table in **Appendix G.7**, demonstrate that CYCle preserves collaborative fairness and reward alignment even in larger populations, while matching the average performance of Gossip-SGD.
>
> We also discuss how CYCle can be adapted to partial participation by operating on induced subgraphs over active clients and computing local reputations without requiring global coordination. While our focus remains on cross-silo settings, we outline the extension strategy in the paper.
>
> These additions are included in the revised manuscript under **Appendix G.7 (titled Scalability to Larger Populations)**, with all relevant changes highlighted in blue for ease of reference.
>
> **W4. Extent of theoretical claims.** We agree with the reviewer that the theoretical analysis in Appendix A is restricted to a specific setting, two-client mean estimation, and is not intended to serve as a general theoretical guarantee for arbitrary learning problems or neural network models. The goal of this analysis is to illustrate the fundamental conditions under which standard methods can lead to negative collaboration gain, thereby motivating the need for similarity-aware collaboration as implemented in CYCle.
>
> To avoid any misunderstandings about the generality of this result, we have now **explicitly clarified these limitations in Appendix A** of the revised manuscript.

---

> ### Author Response · Authors · 2025-06-04
> **Official Comment by Authors (2/2)**
>
> **SC. Communication cost.** Thanks for the suggestion. The details are now reported in Appendix E (titled Computational and Communication Complexity).
>
> **M1. Typos.** Thank you for carefully checking the proof. We would like to acknowledge the typo in reference to Eq. (14); it should indeed refer to Eq. (16) instead, and we have corrected this in the revised version.
>
> Regarding the second line of Eq. (17), we confirm that it is not a derivation step but a repetition of the previous expression, kept intentionally to maintain clarity for the subsequent step, where we apply an upper bound and drop the mentioned term. Thank you again for your attention to detail.
>
> **M2. Notations.** Thanks for this helpful suggestion. We are glad to hear that the notation was generally clear, and we appreciate the feedback regarding its use in later sections. To improve clarity and readability throughout the paper, we added a notation table in Appendix D summarizing all key symbols and their meanings for quick reference.
>
> **OSQ1. Restate the advantages.** We agree that clearly reiterating the motivation and benefits of our approach at this point would help orient the reader before diving into formalism. In the revised manuscript, we have added a brief paragraph just before Section 2.1 that summarizes how CYCle differs from prior work.
>
> **OSQ2. Suggestion on experimental setting.** This is a great suggestion. DomainNet and similar datasets with strong distributional shifts are indeed ideal testbeds for evaluating the selectivity and specialization capabilities of our protocol. In these cases, our similarity-based reward mechanism would naturally promote stronger collaboration within-domain, while suppressing harmful cross-domain interference, thereby reinforcing fair and effective sharing. While we did not include DomainNet, we included a similar real-world dataset, Fed-ISIC2019. We acknowledge this direction and plan to investigate it in future work with an exciting idea in the following point.
>
> **OSQ3. Connection to PFL.** Absolutely, this is an exciting line of thought. The reputation matrix in CYCle can indeed be interpreted as a weighted adjacency matrix, implicitly forming a client-client collaboration graph based on objective similarity. In that sense, the protocol can be extended to identify clusters or cliques of clients that share similar tasks, allowing for more structured forms of clustered or graph-based personalization. We did not explore this formally in the current work, but we will thoroughly explore this perspective and its potential as a direction for future research.
>
> **OSQ4. Intuitions about extension to other tasks?** We believe the core principles of CYCle naturally extend to NLP tasks, especially in settings with user-specific language data or domain-adapted models (e.g., federated fine-tuning of LLMs or adapters). In such cases, the heterogeneity in label distributions or linguistic patterns would manifest in gradient dissimilarity, allowing CYCle to promote alignment among compatible clients. We expect the method to be particularly effective in privacy-preserving personalization of LLMs.

---

> > ### Comment · Reviewer_5Hrc · 2025-06-12
> > **Rebuttal Acknowledgment**
> >
> > I thank the authors for their complete rebuttal, for having substantially revised the placement of contents, and having clarified the extend of the theoretical claims. I acknowledge that the revised paper solves my initial concerns.
> >
> > Regarding W4, I would appreciate further discussion about advancing the theoretical guarantees of collaborative fairness. I belive other researches interested in similar topics could significantly benefit from author's insights on the difficulties of providing more general guarantees not tied to a specific problem or number of clients.

---

> > > ### Author Response · Authors · 2025-06-12
> > >
> > > We thank the reviewer for the acknowledgement of our rebuttal.
> > >
> > > Regarding the question raised, we would like to refer the reviewer to the **last paragraph on page 20**, where we discuss technical challenges for extending the analysis.

---

### Author Response · Authors · 2025-06-04

We sincerely thank all three reviewers, Reviewer XGxa, Reviewer xUX9, and Reviewer 5Hrc, for their thoughtful and constructive feedback. We are especially grateful for the strong alignment across reviews in recognizing the core strengths of our work. Below, we summarize the points:
- **Methodological Innovation and Problem Significance.** All reviewers acknowledged the importance and novelty of our approach to fairness in decentralized learning. Reviewer XGxa highlighted our introduction of gradient alignment into reputation scoring as a key innovation. Reviewer xUX9 and Reviewer 5Hrc both emphasized that fairness is an underexplored yet critical challenge in decentralized learning, and noted that our work provides a principled and clean reformulation of the traditional risk minimization objective in federated learning.
- **Theoretical Rigor and Conceptual Clarity.** All three reviewers (XGxa, xUX9, and 5Hrc) appreciated the inclusion of a formal theoretical analysis that strengthens the credibility of our approach. Reviewer 5Hrc in particular noted the value of the lower bound that compares favorably against FedAvg in terms of fairness, while XGxa and xUX9 acknowledged the conceptual soundness and clarity of the proposed fairness objective and its theoretical grounding.
- **Empirical Comprehensiveness and Robustness.** Reviewers XGxa and xUX9 both praised the depth of our experimental validation, including evaluations on CIFAR-10, CIFAR-100, and Fed-ISIC2019, under various heterogeneity and imbalance conditions. Reviewer 5Hrc also remarked that the empirical results effectively validate our core claims, particularly how CYCle learns to identify and balance contributions in a decentralized setup.
- **Reproducibility and Transparency.** Reviewer XGxa specifically appreciated our detailed reporting of hyperparameters and setup, noting that it provides a valuable benchmark for future research in this area.

We are grateful for these affirmations of our contribution and have incorporated all suggestions for clarification and improvement in the revised manuscript, with relevant updates clearly highlighted in blue. Thank you again for helping us improve the clarity. Individual reviewer concerns are addressed in detailed responses.

---

### Decision · Action_Editor_Xa6o · 2025-07-11

**Recommendation:** Accept as is

**Audience:**

Yes

**Audience Explanation:**

Reviewers were also unanimous on this. Multiple noted that the problem is clearly of interest, and that the method would inspire some amount of interest from related research. Reviewers also noted that the experimental rigor and transparency about things like hyperparameters helps elevate the chance that the method can serve as a benchmark for related work. Moreover, the limitations of the theory aside, the reviewers' interest in extensions to more general settings I think demonstrates that there is an appetite for this kind of work on the theoretical side as well.

**Claims And Evidence:**

Yes

**Claims Explanation:**

The reviewers are quite unanimous: the paper tackles an important problem (collaborative fairness in distributed learning), and develop a worthwhile method for enhancing collaborative fairness. While reviewers had minor qualms with the limitations of the theoretical setting in this work (such as its limitation to two-client mean estimation problems). Reviewers were especially positive about the paper's experimental evidence for the efficacy of the method. Reviewers were also appreciative of the changes the authors made to clarify the scope of the work (such as its theory and relation to other works), which aids in my confidence that the the work directly supports its core assertions regarding the efficacy of and intuition behind the method.